# Social clustering of preference for female genital mutilation/cutting in south-central Ethiopia

**Sarah Myers** [1,2] ✉, **Eshetu Gurmu**[3], **Alexandra Alvergne** [4], **Daniel Redhead** [5,6,7], **Janet A. Howard**[1] & **Mhairi A. Gibson** [1] ✉

Recent estimates indicate that half of Ethiopian girls aged 15–19 years have experienced female genital mutilation/cutting (FGMC). Establishing whether and how pro-FGMC norms are maintained through social transmission is a key priority for global eradication efforts. Here we present the first large-scale socio-centric social network study estimating social influence and social selection on preference for cutting female relatives using data from 5,163 Ethiopian Arsi Oromo adults. Statistical modelling, which accounts for network dependence in cross-sectional data, finds signals of 'contagion' within chatting, respect and money-borrowing networks. This indicates that social influence contributes to FGMC maintenance. We find no clear evidence of social selection within marriage-advice networks, suggesting these networks are not implicated in FGMC maintenance. Contrary to assumptions underpinning current eradication efforts, we find negligible evidence that FGMC is a social coordination norm (with only 6.3% endorsement) or an empirical or normative expectation. We conclude by making intervention design recommendations.

Female genital mutilation/cutting (FGMC) is estimated to affect over 200 million women and girls alive today, occurring within more than 90 countries across Africa, the Middle East and Asia, and within diaspora populations globally[1]. Ethiopia has one of the largest populations of women and girls to have experienced FGMC[2] and the tenth highest rate in Africa[3]. Though the practice was outlawed in 2005, almost half of Ethiopian girls aged 15–19 years were estimated to have been cut in 2016[4]. There is substantial variation in the age at which cutting occurs by region; in Oromia, where our study is located, 50% of cut 20–24 year olds underwent the procedure before the age of 10 years and 18% at or after age 15 years[3]. While establishing why FGMC persists is a priority

for global policymakers, how FGMC norms are socially transmitted and maintained remains unclear. There is currently a dearth of empirical data regarding FGMC with which to test prevalent ideas within the literature and there are growing calls to urgently address this deficit[5], as existing intervention strategies appear inadequate to meet the United Nations Sustainable Development Goal of eliminating the practice by 2030[6]; for example, while progress has been made in Ethiopia, it needs to be eight times faster than in the previous 15 years to meet this target[2]. In particular, there have been recent calls for the mapping of FGMC norms and behaviour within social networks to inform intervention design[5]. Social network data can enhance understanding of the

[1]Department of Anthropology and Archaeology, University of Bristol, Bristol, UK. [2]BirthRites Lise Meitner Research Group, Max Planck Institute for Evolutionary Anthropology, Leipzig, Germany. [3]Center for Population Studies and Institute of Development and Policy Research, Addis Ababa University, Addis Ababa, Ethiopia. [4]Institute of Evolutionary Science Montpellier, University of Montpellier, Montpellier, France. [5]Department of Sociology, University of Groningen, Groningen, The Netherlands. [6]Inter-University Center for Social Science Theory and Methodology, University of Groningen, Groningen, The Netherlands. [7]Department of Human Behavior, Ecology and Culture, Max Planck Institute for Evolutionary Anthropology, Leipzig, Germany. ✉e-mail: sarah_myers@eva.mpg.de; mhairi.gibson@bristol.ac.uk

distribution of FGMC preferences and practices within communities and identify whether key people, network characteristics or particular types of social relationship are associated with their transmission and maintenance. Here, we combine social network data with insights from cultural evolutionary theory to explore the social transmission of pro-FGMC preference among Arsi Oromo agropastoralists in Ethiopia.

From a cultural evolutionary perspective, pro-FGMC preference is a cultural variant 'copied' or acquired through social learning. Here, social learning is defined as learning that is aided by observation of, or interaction with, another individual or their creations[7]. The distribution of pro-FGMC preference observed within social networks may result from one of two processes, which may or may not be mutually exclusive: (1) social influence, that is, the phenomenon of individuals becoming more similar to their social connections via various mechanisms after a connection is made[8–10] or (2) social selection, that is, the phenomenon of individuals selecting social connections based on homophily (that is, their having traits in common)[11–13]. From an intervention perspective, social influence implies anyone exposed to pro-FGMC preference via their social connections is potentially at risk of becoming pro-FGMC themselves (indeed social influence is often referred to as 'contagion'[10,14]), which may require widespread intervention to prevent transmission, depending on prevalence and the dynamics driving contagion. Social selection implies the need to target specific clusters of pro-FGMC individuals, who may engage in the practice themselves and whose continued presence poses the risk of later contagion. Pro-FGMC preference may differentially transmit through, or form the basis of, many different types of social relationship or ties. Improved understanding of these complex dynamics is of benefit to intervention policies that focus on ending the practice.

Among the Arsi Oromo, FGMC involves a cut or nick to the clitoris and occurs about a month before marriage, once a marriage agreement is in place. In Afaan Oromoo, FGMC is referred to as 'huuba irraa fuudhuu', which can be translated as 'removing the garbage/unwanted'[15]. FGMC was a prerequisite for marriage; however, in recent years, prevalence rates seem to have been falling and remaining uncut no longer prohibits marriage[16]. Of married women interviewed in 2010 in the same population, 90% reported they had been cut; however, when asked about their married daughters, mothers reported that 87% of daughters aged 35 years or over had been cut compared with 53% of those under 25 years. More recently, data collected in 2016 using indirect questioning methods estimated 22% of the population held pro-FGMC preference[17]. This pattern of mixed views and behaviour provides scope for exploring potential differences in the social network dynamics of individuals who report being pro- versus anti-FGMC. Such exploration can enhance our understanding of what may help maintain the practice in the face of its longstanding illegality and national eradication efforts[18]. We focus on FGMC preference, here measured as whether an individual would want FGMC for a hypothetical daughter and/or daughter-in-law; theoretically, favouring either is likely to be important in this context where parents are influential in marriage decisions and both parents and prospective parents-in-law contribute to the cutting ceremony[16].

We present data collected from 5,163 Arsi Oromo adults in 2021–2022, residing in nine neighbouring administrative kebele-zones (Fig. 1a), as part of a cross-sectional study exploring preferences and expectations surrounding FGMC. While FGMC is practiced across Oromia, and elsewhere in Ethiopia, our study site is chosen due to members of our team having worked for multiple decades in the region, developing strong local ties that facilitate the study of such a sensitive topic. Given the illegality of the practice, we purposefully anonymize the site to protect our participants; however, it is in the Histosa Woreda, of the Arsi zone. The site is rural, with ethnically homogeneous agropastoralist inhabitants, predominantly practising subsistence farming of maize and wheat and limited cattle herding. Most households lack electricity and running water, education is increasing but still rare beyond primary

school[16], and Islam is the prevailing religion. Following an initial household census (the Household Census) to identify resident adults, a subsequent survey (the Norms and Networks Survey) was used to gather our focal data, with all respondents posed FGMC-related questions and, at the same time, a random subsample posed additional social network questions (see the Methods for details). This resulted in the collection of network data reflecting four different types of social relationship from 2,545 individuals: who people spent time chatting with (the chatting network), who they respected and admired (the respect network), who they can borrow money from (the money-borrowing network) and who they would go to for advice regarding preparing a daughter for marriage (the marriage-advice network). This allows us to explore the distribution of pro-FGMC preference across a range of estimated socio-centric networks.

Drawing inferences of cultural/norm transmission from cross-sectional data must be done with caution and we acknowledge this limitation from the outset; however, we employ a range of recent statistical techniques to mitigate many historical issues with social network analysis and maximize our inferential power. First, when using double-sampled name generators (that is, networks elicited by the combination of responses to questions tapping both sides of the relationship in question, for example, 'Who do you give to?' and 'Who gives to you?'), as in the money-borrowing and marriage-advice networks, we employ a technique that improves the estimation of the latent network[19,20]. Second, social network studies have been plagued by failure to account for the network dependence of individual outcomes[10], potentially confounding findings. Here, we take advantage of a new generation of social influence models developed to account for the network dependence in cross-sectional data. Nevertheless, statistically distinguishing between social influence and social selection is not possible without longitudinal data[13], and both may act in concert, therefore our analytical strategy necessarily requires some opening assumptions[21], which in turn inform our modelling decisions and results interpretation.

We assume that if pro-FGMC preference clusters within the chatting, respect or money-borrowing networks then social influence will be the most important cause. This is based on the fact that these are relationships influenced by a multitude of factors, such as shared interests, age, education and social position, to name a few (for example, refs. [22–26]) and it seems unlikely that FGMC preference would reliably rank highly amongst this multitude. We use auto logistic actor-attribute models (ALAAMs)[10,27], which were developed to explore signals of social influence or 'contagion' within cross-sectional network data, to predict an individual's likelihood of pro-FGMC preference on the basis of the presence of pro-FGMC preference among their social ties within the chatting, respect and money-borrowing networks. We use a recent Bayesian implementation[10]; new advances have also been made to Frequentist implementations[28]. Our primary interest is in the potential for 'direct contagion', that is, social influence occurring between individuals who interact directly with one another; however, numerous forms of contagion have been proposed (for accessible, nonexhaustive overviews see refs. [27,29]) and we explore the potential for several other forms of contagion post hoc.

However, given the close association between FGMC and marriage in Arsi Oromo culture[16], it does seem plausible that social selection based on shared FGMC preference would guide who individuals would seek advice from regarding preparing their daughter for marriage. Further, the ties elicited regarding such advice were hypothetical in nature for a large portion of participants who were yet to have/never had a marriable daughter (the median age of the sample is 30 years and 75.3% do not have a married daughter). Where this is the case, social influence is an unlikely cause of pro-FGMC clustering (that is, one cannot be influenced by advice that was never received). As such, we employ combined stochastic block and social relations models[20] designed to explore signals of social selection within the marriage-advice network,

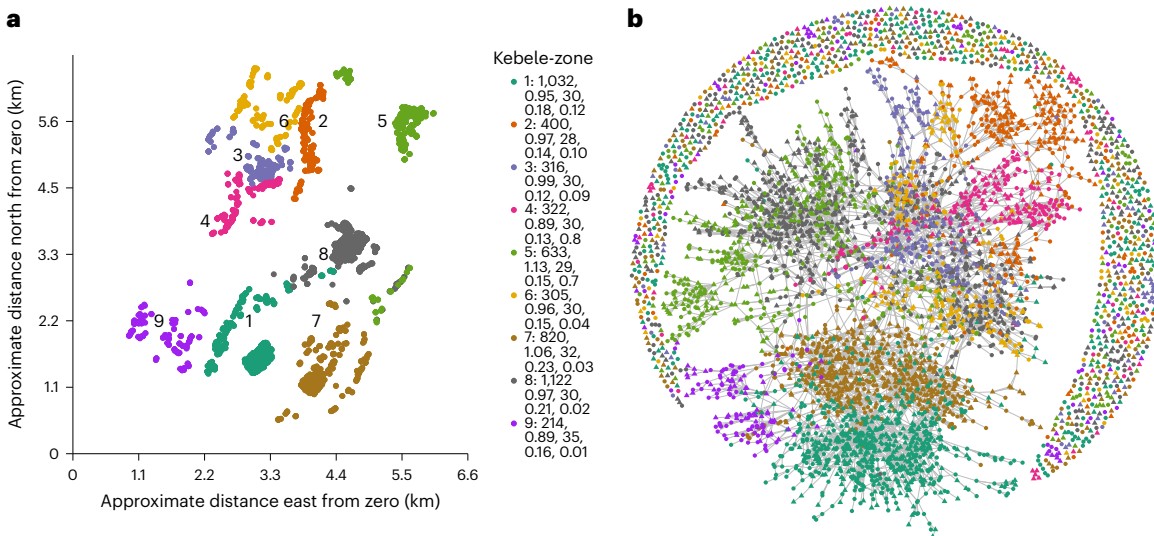

**Fig. 1 | Visualizing social connections. a**, A map of the relative geographic positions of households belonging to respondents to the Norms and Networks Survey. The legend indicates the number of each kebele-zone followed by the sample size of respondents, gender ratio, median age, proportion with some secondary education or beyond and proportion reporting pro-FGMC preference. The plot is missing the household locations of 51 respondents not censused in the Household Census. **b**, A plot of chatting ties between 5,163 respondents to the Norms and Networks Survey as a whole, reported by the 2,545 respondents interviewed for network data. Plotted using the Fruchterman–Reingold algorithm, ties can be seen to cluster by kebele-zone, with only limited ties between individuals in different kebele-zones. Not all respondents to the survey were named as chatting partners (n = 1,071) (including 27 network respondents who both named an entirely unknown network and were not themselves nominated), hence the presence of isolated nodes at the outer rim.

by predicting the likelihood of a tie between any two members of the network based on their FGMC preference (dis)similarity.

Finally, the public health literature on FGMC has been heavily influenced by social norms theories, though so far empirical testing remains limited[5]. The social norms literature is wide ranging, with numerous competing and conflicting concepts and overlapping terminology, having been influenced by multiple disciplines[30]. However, there is a consensus that norms are social and mutually held by some members of a group, they are tied to actions and influence decision-making and they can be both prescriptive (encouraging) and proscriptive (discouraging). Within the field of cultural evolution, individuals are seen as enacting social learning strategies, which shape what is copied, when it is copied and who it is copied from. Social learning strategies act as selective cultural evolutionary forces acting on the frequencies of cultural variants within a population[31–33]. A social learning strategy may be to simply copy randomly or copying may be biased in a multitude of ways[33], with cultural variants transmitting via different paths through networked communities and leaving different frequency signals in a population[34,35], dependent on the bias. Debates as to the role of social learning strategies in norm psychology is ongoing[36]; nevertheless, as norms entail conformist behaviour underpinned by varying motivations, differing social norm formulations will also leave distinct frequency signals, which may be used to assess claims of a specific norm's explanatory power (for example, ref. [37]). As such, if social influence is the phenomenon of becoming more similar to one's social connections, then social learning strategies and norms will shape patterns of social influence observed within socio-centric networks. We also present descriptive data that explore two dominant social norm narratives in the literature used to explain the maintenance of FGMC. One, that FGMC is a social coordination norm tied to marriageability, to which almost all members of a community are expected to conform to ensure the social prospects of their daughters[38]. To infer the applicability of this explanation, we explore the prevalence of pro-FGMC preference across kebele-zones. The second, that pro-FGMC preference either constitutes a 'descriptive norm', persisting because people are motivated by an empirical expectation that others in the group are in favour of it, or a social norm stemming from a combination of both an empirical

expectation and a normative expectation that others approve of it and so there are anticipated social costs to not participating[39,40]. Here, we harness answers to questions regarding respondents' expectations of other kebele-zone members preferences and approval of FGMC. Each narrative has differing implications for intervention strategies, to which we return in the discussion.

## Results

### FGMC preferences and norm expectations across kebele-zones

The geographic distribution of respondent households across the nine kebele-zones, spread across an area roughly 5 km², along with zone-level demographic characteristics can be seen in Fig. 1a. The age make-up of respondents across kebele-zones was similar, ranging from a median of 28 to 35 years, while the gender ratio of men to women ranged from 0.89–1.13. In every kebele-zone, about half of respondents had received only some primary school education, ranging from 42.8% to 50.3%, while the percentage of those receiving some secondary education or beyond ranged from 11.7% to 22.7%. In terms of religion and ethnicity, 94.1% of the sample was Muslim, which in this region is indicative of Arsi Oromo ethnicity, while 5.8% were Orthodox Christian and four individuals were of other religion. Of those who were Orthodox, 77.3% lived in kebele-zone 7. For a full breakdown of measures by kebele-zone see Supplementary Table 1 (note that we do not provide confidence intervals for these statistics as our sample approximates the full adult population).

At the sample level (n = 5,163), respondents overwhelmingly reported not wanting FGMC for a daughter/daughter-in-law, with only 6.3% self-reporting yes to wanting FGMC; throughout the following we use the terms pro- and anti-FGMC as shorthand to indicate those who reported yes and no, respectively, to wanting FGMC for a daughter/daughter-in-law. Of those categorized as pro-FGMC, 52.3% reported wanting FGMC in relation to both daughters and daughters-in-law, 34.8% in relation to daughters-in-law only and 12.9% in relation to daughters only. FGMC preference was not strongly stratified by demographic characteristics (Supplementary Fig. 2), though being pro-FGMC was somewhat more common at older ages. Pro-FGMC

preference was also held by both Muslim and Orthodox individuals, with 6.5% and 2.7% pro-FGMC prevalence, respectively.

The social coordination norm argument[38], informing many interventions, assumes FGMC support is maintained at a high prevalence (because it grants access to marriage); however, across kebele-zones, pro-FGMC preference ranged from 1.4% to 12.1%. Variance in the prevalence of pro-FGMC preference across kebele-zones is not clearly associated with zone demographics, as is evident when comparing the two largest zones (zone 1 $n = 1,031$ and zone 8 $n = 1,122$): each has slightly more women (gender ratio 0.95 and 0.97, respectively), has a median age of 30 years, differ by 1.9% in the representation of uneducated individuals (24.4% versus 22.5%) and 3.3% in representation of those secondary educated or above (17.6% versus 20.9%), and yet 12.1% of zone 1 report pro-FGMC preference compared with 2.3% of zone 8.

We find no compelling evidence that pro-FGMC preference is maintained by a 'descriptive norm'. When exploring the empirical expectations that respondents have regarding pro-FGMC preference in their kebele-zone, respondents reported broadly that they thought the percentage of both men and women in their zone wanting FGMC for their daughter was low, with the majority stating no men (74.7%) or women (79.4%) would want it. While those with pro-FGMC preference were more likely to think at least some proportion of zone members would want FGMC as compared with those with anti-FGMC preference (31.1% versus 77.6% stating no men would want it and 42.2% versus 81.9% stating no women would), if above zero, they typically did not think the proportion was high: 46.8% stated that 10% of men would want it and 4% stated 50% or more men would, while the equivalent percentages were 32.9% and 6.2% for women.

When broken down by zone, patterns suggest that pro-FGMC respondents are more aware of the views of other zone members than those anti-FGMC (Supplementary Table 2 and Supplementary Fig. 3): while the majority of anti-individuals always report there is 0% support for FGMC, pro-individuals are broadly more likely to report the decile closest to the zone-level reported prevalence, that is, either 0% or 10%.

To meet the criteria of a social norm as defined by Biccheiri[40], pro-FGMC preference must be influenced by both an empirical expectation and a normative expectation that others approve of cutting. At the sample level, 0.5% thought that others in their kebele-zone would approve if a local family arranged for their daughter to be cut, while 9.8% thought others would think it was none of their business, and 89.7% thought others would disapprove. In terms of differing normative expectations, those holding pro-FGMC preference were proportionally more likely to report others would approve (3.1% versus 0.4%) or think it was none of their business (19.1% versus 9.1%), and less likely to report others would disapprove (77.8% versus 90.5%) compared with those anti-FGMC. This pattern was broadly repeated across zones (Supplementary Table 3). However, as only 3.1% of pro-FGMC individuals reported that 10% or more men or women in their zone would want FGMC and others would approve of FGMC arrangement, the argument of Biccheiri and Marini[39] that FGMC is maintained by a combination of perceived empirical dominance and social approval is not supported in this context.

## Characterizing Arsi Oromo social networks

The four different social networks we collected appear predominantly distinct from one another; Jaccard similarity coefficients for the alters named in response to all combinations of name generator show only small effect sizes (Fig. 2a). This increases confidence that, for example, a positive signal of contagion in the respect and chatting networks is not due to the same relational ties in both. This is despite high numbers of kin being nominated across networks (Fig. 2b). In terms of the relationship types reflected in alter nominations, the respect network had the highest proportion of non-kin nominations, while at least 30% of nominations went to alters defined as unrelated friends or neighbours in response to all but the marriage-advice name generators. Women made more nominations to relatives by marriage than men and

consistently made a higher proportion of nominations to unrelated friends or neighbours. Among men, brother was the most commonly used kin category, in all but the respect network, and alternated with unrelated friends or neighbours across networks as the most commonly used category overall. There appears to be no stark differences in the composition of networks between those reporting pro- versus anti-FGMC preference; pro-FGMC individuals' have slightly more kin-dominated money-borrowing networks, and anti-FGMC individuals slightly more kin-dominated chatting networks (Fig. 2b), while the distribution across relationship type at a more granular level shows no clear patterns (Supplementary Fig. 4). Finally, the analytical subsets are broadly representative of respondents' full reported networks (Supplementary Table 4) (for further discussion, see the Supplementary Information).

The structural characteristics of the different networks can be seen in Table 1 and the plotted networks in Fig. 1b and Supplementary Figs. 5–7. Broadly speaking, while the characteristics of the estimated networks differ across network types, for each type of network, there is a negligible difference between the network characteristics of those reporting pro- versus anti-FGMC preference (Supplementary Table 5). Similarly, the prevalence of pro-FGMC preference does not seem to be associated with kebele-zone-specific characteristics of the networks (Supplementary Table 6). These network characteristics are calculated from identified alters who participated in the Norms and Networks Survey; for a comparison of the latent money-borrowing and marriage-advice networks estimated using the analytical subset of respondents who reported their networks, see Supplementary Table 7.

## Social influence model results

We investigated social influence by exploring signals of 'contagion effects' using a Bayesian implementation of ALAAM models, which predict an individual's likelihood of pro-FGMC preference on the basis of the presence of pro-FGMC preference among their social ties. These models are adjusted for age, gender, education, zone and the number of nominations made (out degree) and received (in degree). We found a compelling signal of contagion across the chatting, respect and money-borrowing networks. This is indicated by the posterior distribution of estimates for the 'contagion' parameter specified in our models, plotted in Fig. 3. When assessing all alters, in each network the bulk of the posterior distributions fall above zero, indicating that having a tie with at least one other pro-FGMC individual is positively associated with the probability of an individual holding pro-FGMC preference themselves. There is also reasonable evidence for direct contagion occurring across both kin and non-kin ties, though the effect is broadly indicated to be slightly smaller among kin ties. Post hoc exploration of additional forms of contagion indicates that these direct contagion signals are broadly robust to the inclusion of alternative dependencies, which themselves typically have wider posteriors straddling zero. The contagion signal among kin nominated within respect networks is an exception and possibly reflects a signal of indirect contagion (that is, social influence stemming from indirect, rather than direct, ties to pro-FGMC preference holders) (Supplementary Fig. 8); however, we have no strong theoretical explanations for such an effect, and thus the process through which this form of contagion may operate is unclear. Full ALAAM results of the direct contagion models, including models using latent money-borrowing networks estimated from only those who reported networks, can be found in Supplementary Tables 8–10 and Supplementary Fig. 9. The results of goodness-of-fit simulations can be found in Supplementary Tables 11–19, the full results of the 'best fitting' post hoc models can be found in Supplementary Tables 20–22 and the posteriors for the contagion parameters are plotted in Supplementary Fig. 8.

## Social selection model results

To detect signals of social selection or homophily in the marriage-advice network we ran combined stochastic block and social relations models

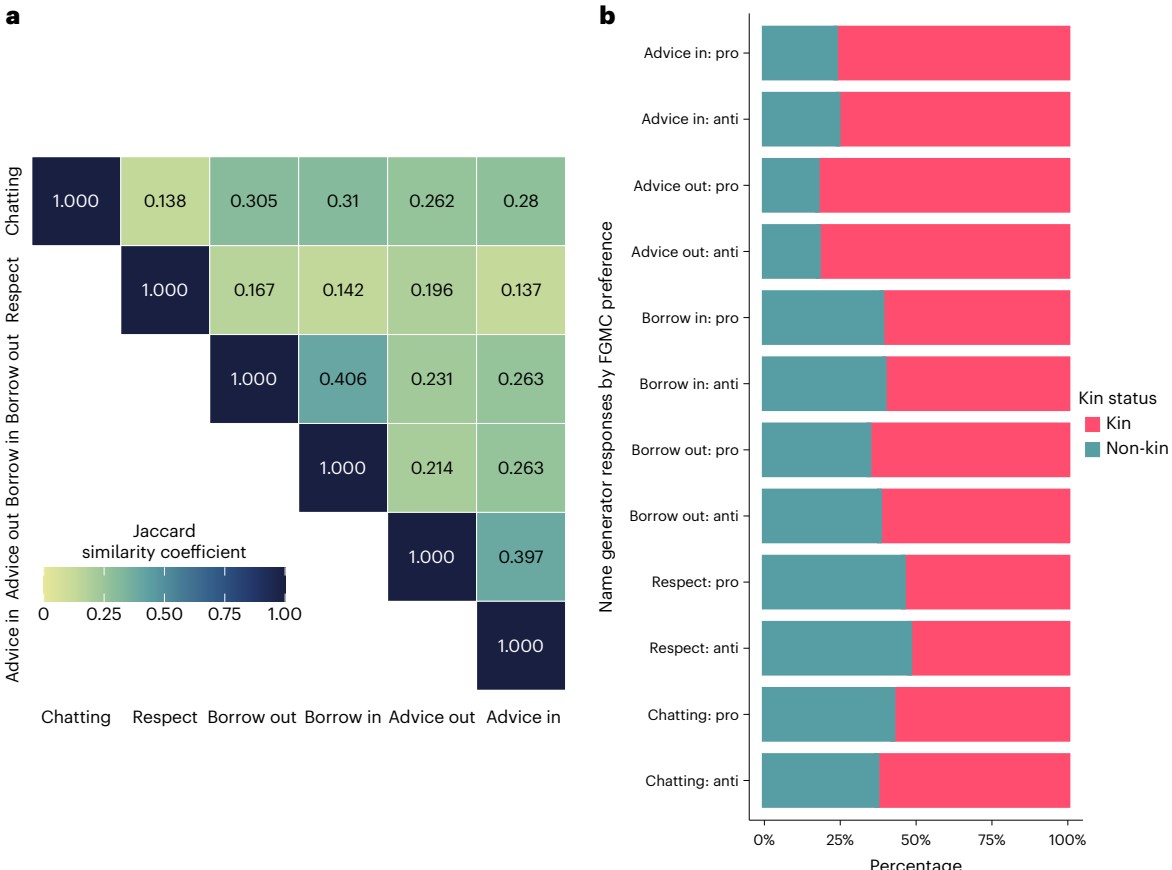

**Fig. 2 | Social network descriptive statistics. a**, A matrix of Jaccard similarity coefficients indicating the degree of overlap between identified alters elicited across name generators (the darker the cell, the higher the overlap). **b**, The percentage distribution of kin (pink, right), that is, relatives, versus non-kin (teal, left) nominations across name generators dependent on whether the nominator reported holding pro- or anti-FGMC preference. Note the 'in' and 'out' attached to advice and borrow indicate the double sampling used to elicit the marriage-advice and money-borrowing networks, respectively.

**Table 1 | Structural characteristics at the network level**

| Characteristic | Chatting | Respect | Money | Advice |
|---|---|---|---|---|
| Ties | 7,462 | 6,804 | 8,733 | 9,021 |
| Density | 0.0003 | 0.0003 | 0.0003 | 0.0003 |
| Reciprocity | 0.183 | 0.029 | 0.422 | 0.442 |
| Transitivity | 0.203 | 0.067 | 0.121 | 0.205 |
| Isolates | 1,071 | 1,862 | 1,201 | 1,346 |

The number of ties reflects the total number of relationships held with the 5,163 survey respondents, reported by the 2,545 network interviewees. Density reflects the ratio of actual ties reported to possible ties and measures the connectivity of the network. Reciprocity reflects the proportion relationships which are mutually reported, possibly indicating the cohesiveness present in the network (though note this estimate is biased downwards by the inability of a portion of the sample to report their ties). Transitivity reflects the tendency for closed relationship triangles within the network, indicating the interconnectedness of the network (this will be similarly biased downwards). The number of isolates reflects the number of individuals who were either not named in response to the name generator or reported an entirely unidentified network.

using Bayesian estimation to predict the likelihood of an advice tie between two individuals based on shared FGMC preference. These models are adjusted for age, gender, education, community role and ties in the chatting, respect and latent money-borrowing networks. The results do not show a compelling signal because shared FGMC preference is not associated with the likelihood of a marriage-advice tie between two individuals. As can be seen in Fig. 4a, in each kebele-zone all offset estimates heavily overlap, indicating no preference combination is more likely than another; as such, the lower probability of anti-FGMC individuals nominating pros, compared with fellow antis, in zones 7 and 9 (Fig. 4b) should be treated with scepticism. Full model results can be seen across Supplementary Figs. 10–14 and Supplementary Tables 23–31.

## Discussion

We sought to identify whether indications of social influence or social selection in relation to pro-FGMC preference (as defined by the desire to have a hypothetical daughter or daughter-in-law cut) could be found within different types of social networks among this Arsi Oromo sample. We find clear signals of contagion, suggesting that pro-FGMC preference is transmitting/maintained via social influence within chatting, respect and money-borrowing networks among this sample of adults. Direct contagion signals were similar for both kin and non-kin, which suggests that relatives are not primary in FGMC preference formation. We find no clear evidence of social selection in hypothetical marriage-advice networks, suggesting individuals do not preferentially seek advice regarding the marriage of daughters from those who hold the same FGMC preferences as them. Further, with low and locally variable prevalence, pro-FGMC preference does not appear to be a social coordination norm. Nor do the majority of those holding pro-FGMC preference appear to expect others in their kebele-zones to also want cutting or to approve of its being performed. From an intervention perspective then, these results indicate that prespecified, cross-culturally generalizable patterns of norm transmission should not be assumed.

Our results suggest multiple forms of social relationship through which pro-FGMC preference may persist via social influence and,

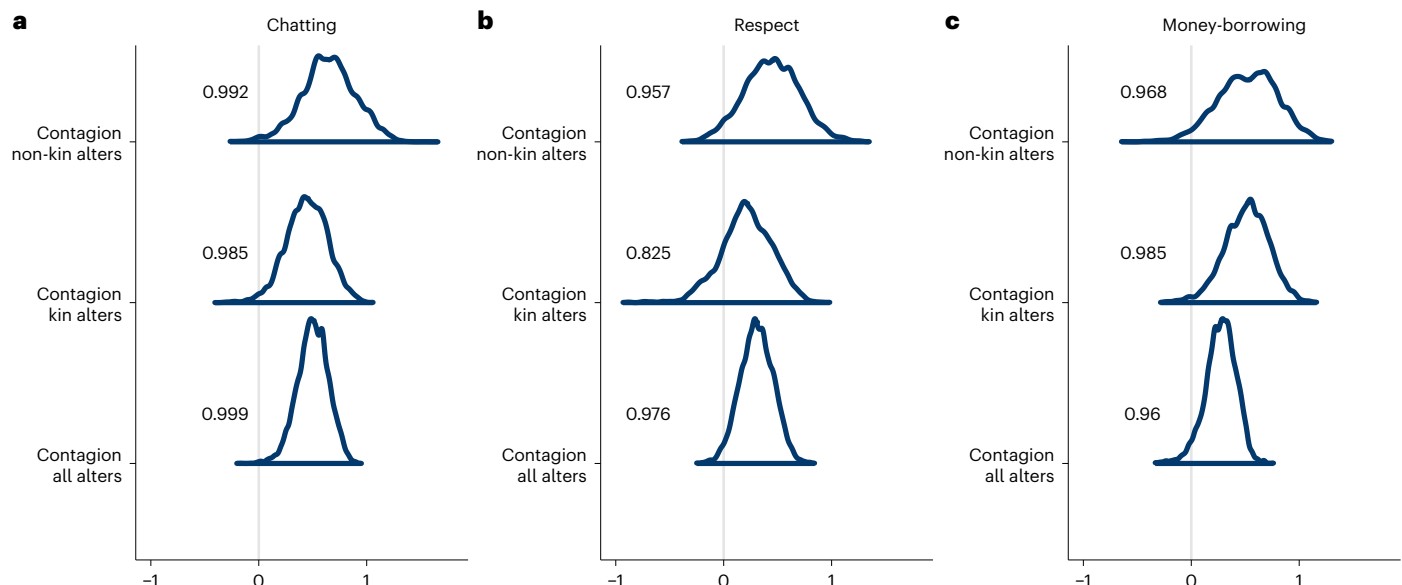

**Fig. 3 | Social influence signals across networks. a–c**, Plotted are the posterior distributions for the estimates for social contagion from ALAAM models assessing either all, kin only or non-kin only alters within the chatting (**a**), respect (**b**) and latent money-borrowing (**c**) networks. Estimates above zero signal a positive contagion effect; please note the differing *x*-axis scales. The numbers listed within the graphs give the proportion of the posterior above the null (that is, zero).

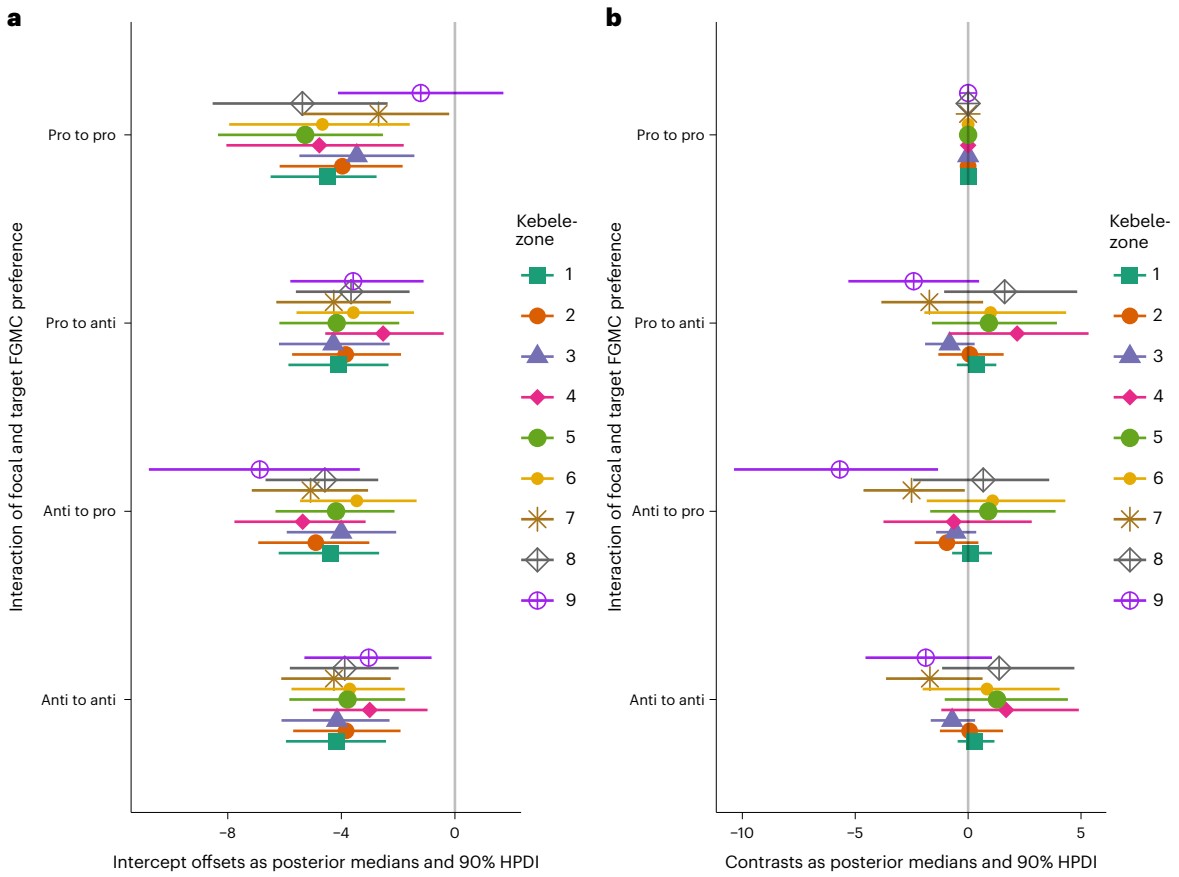

**Fig. 4 | No signals of social selection in marriage-advice networks.**
**a,b**, Plots of the combined stochastic block and social relations model estimates of the log-odds of a tie of a given FGMC preference combination offset against a global intercept term (**a**), from which contrast coefficients (**b**) are calculated. Nonoverlapping offset estimates across preference combinations can generally be considered to be consistently different effects, while contrasts quantify the difference by tethering estimates to a specific reference preference combination (here pro to pro; for alternative reference categories see Supplementary Fig. 9). As such, there is negligible evidence of FGMC preference homophily among ties. Here, 'focal' indicates the seeker of advice and 'target' indicates to whom they would go. The point estimate reflects the median and the error bars the highest posterior density interval (HPDI) from the posterior distribution. For the number of advice out and advice in nominations contributing to the latent networks per kebele-zone, see Supplementary Table 33.

potentially, increase in frequency de novo. The results were similar when exploring direct contagion in each of the estimated chatting, respect and money-borrowing networks, and these networks showed only minimal overlap. While the rate of openly expressed pro-FGMC preference was low, 6.3% at the sample level, which may be taken as an indicator of the success of eradication efforts, this figure probably represents an underestimate resulting from direct questioning. Previous work within this community suggests this leads to at least 10% of respondents masking their true attitudes[17]. However, it is unknown whether there are local dynamics influencing the likelihood of under-reporting, thus whether differences at the kebele-zone level (ranging from 1.4% to 12.1%) reflect underlying variation in attitudes. While it was estimated that 46% of adolescent girls aged 15–19 years living in Oromia had been cut in 2016[3], this estimate encompasses a large geographic area and multiple ethnic groups; the most recent local estimate dates from 2010 when 53% of married women aged 25 years or under had been cut. Unfortunately, we do not know the current prevalence of cutting performed in our study community and so, by extension, the correlation between preference and practice. Nevertheless, even very low rates of pro-FGMC preference should be of concern due to the possibility of contagion, because unanticipated sociocultural changes may spark resurgence—as seems to have been the case in Kenya during the COVID-19 pandemic[41]. Indeed, the potential for such transmission at low FGMC frequency supports the wisdom of the second principle underpinning the United Nations 2030 Agenda for Sustainable Development, 'Leave no one behind'[42].

We find no clear evidence of FGMC social clustering through social selection in hypothetical marriage-advice networks, as people sharing the same preference was not predictive of them sharing an advice tie. This suggests that despite the close association between cutting and marriage in this community, the seeking of advice regarding the marriage of daughters does not provide a context in which FGMC like-minded people preferentially come together. The degree to which this is a consequence of recent declines in pro-FGMC preference or shifts in marriage practices from historically prevalent arranged marriages to increasingly common 'love matches'[16] is unclear. Nevertheless, this is important because, while we cannot rule out pro-FGMC contagion during actual advice exchange, it indicates that marriage-advice networks do not currently support mutually reinforced pockets of pro-FGMC individuals. Marriage advice remains a transmission path those designing interventions should consider, but given the findings from our models of other networks, it should not be an exclusive target.

The data presented here also cast doubt on the relevance of two prevailing social norm narratives within the intervention literature to FGMC in the context of this Arsi Oromo community. Perhaps the most influential approach stems from the work of Mackie[38], whose 'convention hypothesis' contends that FGMC developed within the context of a coordination game in which all parties face incentives (that is, daughters' marriageability versus ostracism) to match strategies, resulting in high levels of conformity. Under this framework, FGMC is seen as a coordinated practice used by families to prepare daughters for marriage and it implies there is a threshold associated with the proportion of families cutting daughters and demanding cut daughters-in-law, which maintains the practice. Shifting enough families below this threshold will then cause the practice to rapidly vanish, because families can secure marriage for their daughters without the health costs of cutting. Public declaration of abandonment interventions, promoted by the United Nations Population Fund–United Nations Children's Fund (UNICEF) Joint Programme on the Elimination of Female Genital Mutilation, have harnessed this logic, with mixed success[43,44]. Game theoretic modelling indicates either very high or very low FGMC acceptance and/or practice are expected under this social norm framework[37]; where prevalence rates exist for prolonged periods at lower, but not extremely low levels, as appears to be the case in this study community, then a social coordination game-based norm is unlikely to be maintaining the

practice. However, how long is 'prolonged' or conversely how rapidly the practice should vanish, has not been formalized, making conclusive testing of the convention hypothesis not yet possible. Nevertheless, our data add to a limited but growing number of empirical studies finding that attitudes and behaviour surrounding FGMC are very variable at small local scales[37,45–48].

A second powerful norm-related perspective is that FGMC is variously attached to moral, religious and gender norms, which may result in it being a social convention[49] or social obligation or expectation[5]. FGMC is thought to constitute either a 'descriptive norm', persisting because people are motivated by an empirical expectation that others in the group are in favour of it or a norm stemming from both an empirical expectation and a normative expectation that others approve of it and so there may be social costs to not adopting the sentiment and behaviour of others[39,40]. While we find pro-FGMC people are more likely than anti-FGMC people to report perceiving at least some other people in their kebele-zone as being pro-FGMC, the majority still thought no more than 10% were in support of the practice—this does not favour empirical expectations underpinning the maintenance of pro-FGMC preference. Similarly, though pro-FGMC people were more likely to report thinking other people in their kebele-zone would approve if a local family arranged FGMC for their daughter, this amounted to only 3% of pro-individuals. This indicates there is also not a normative expectation of pro-FGMC preference maintaining it within this subsection of the Arsi Oromo community. Again, this complicates foundational assumptions informing the perspectives of global public health bodies, for example, UNICEF[49].

Taken together our findings have important implications for intervention strategies relying on a one-size-fits-all approach. The much-vaunted interventions using public declarations as triggers 'tipping' FGMC into collapse rely on the vast majority of the population coordinating to gain some reward (for example, their daughter's marriage) and individuals within the population being homogeneous in terms of their decision-making process regarding what will lead them to decide to cut or not (that is, the same exposure will tip them all)[50]. Given the apparent absence of mass coordination to gain access to marriage for daughters, indicated by low levels of professed pro-FGMC preference and marriage without cutting[16], such intervention seems unlikely to be efficient in Arsi Oromo communities. Further, Efferson et al.[50,51] have recently highlighted the improbability of homogeneity when it comes to tipping points and the problems this poses for applied cultural evolution. Indeed, that people in our sample hold pro-FGMC preferences despite thinking that the majority of others in their zone have 'tipped' into not wanting and/or disapproving of cutting, points to heterogeneity. As such, our study acts as a useful negative example, cautioning against blanket application of interventions based on influential assumptions of mass coordination[43,44].

In addition to finding signals of contagion in multiple types of networks, we also find that among our sample of Arsi Oromo adults, while pro-FGMC preference was more likely among those with no education and older age groups, it was present in all demographic groups (Supplementary Fig. 2). The network position of pro-FGMC people also did not notably differ from those who were anti-FGMC, with measures of average centrality (an indicator of likely control over information transmission) similar irrespective of FGMC preference. This finding indicates both that pro-FGMC people do not hold greater relative importance regarding preference transmission within this community, and that anti-FGMC preference will not replace pro-preference without further shifts in the cost–benefit dynamics influencing preference change. Looked at from one angle, our social network results do not provide a single clear target for policymakers; for example, we find no obvious opinion leaders or demographics to target, and there is the potential for contagion through seemingly many relationships making transmission paths hard to isolate. Nevertheless, we think this is a useful insight. First, social network data are not a silver bullet, while much insight can be gained from their use in relation to public health[52], the

collection of network data is expensive and time consuming and this much be weighed against the additional insights they are expected to provide beyond what can be gained by easier means. Second, in the absence of detailed local data, the potential for social influence acting through multiple relationships across any given community may be a reasonable assumption. This position has the advantage of not precluding FGMC maintenance by mass coordination or social convention, but not assuming them either. From this perspective, steps preventing pro-FGMC contagion on exposure are likely to be more widely effective than actively seeking to convert those favouring cutting.

Interventions that focus more broadly on systemic change to alter the cost–benefit calculations of individuals, to minimize their likelihood of becoming pro-FGMC on exposure to pro-FGMC social contacts (metaphorical 'vaccination') and, conversely, maximize their likelihood of becoming anti-FGMC on exposure to anti-FGMC contacts, may be more successful than trying to 'tip' pro-FGMC individuals en masse. Previous work suggests increasing opportunities for female education and economic productivity among the Arsi Oromo may reduce household pressures to secure early marriage for daughters (facilitated by cutting) and aid parents in abandoning the practice[16]. In our sample, 96.9% and 93.2% of respondents said they wanted their daughter and daughter-in-law respectively to go to college, and 89.8% and 87.1% for them to work in the city; this is currently unlikely for most girls, but interventions focused on facilitating it to become a reality may be both effective and have more far-reaching benefits. An alternative, lower-cost strategy also taking a community-wide approach is 'edutainment'—a communication strategy combining education and entertainment to distribute new ideas and behaviours via mediums such as TV soap operas[50]. The use of edutainment in relation to FGMC (for example, refs. 53,54) and other development-related outcomes[55–57] appears promising.

We have explored whether FGMC preferences socially cluster, as a gateway to drawing inferences about the social processes underpinning the patterning of preferences. This strategy is distinct from exploring the reasons people themselves give for holding the preference that they do, and can bring additional insights. For instance, religion is among the major reasons given by people in Arsi for performing FGMC; however, as can be seen in our data, there is heterogeneity within religious groups, with both pro-FGMC and anti-FGMC preference expressed by Muslim and Orthodox participants. Our results suggest social influence plays a role in generating this within-group heterogeneity.

We have approached our discussion through the lens of our opening assumption that any signal of an effect in our ALAAM results from social influence. While we consider this to be a reasonable assumption, the cross-sectional design of the study means we cannot rule out that some or all of the signal we detected was caused by social selection. Using FGMC preference data collected at the same time as the network data creates an issue of simultaneity and necessitates the assumption that networks are exogenous and stable. While networks are fluid among adults in large-scale populations with high degrees of residential and employment mobility across the life course[58], we feel it is reasonable to assume the networks captured in this small-scale subsistence population are relatively stable. Relatives form a sizeable portion of each network type, residency is patrilocal with men remaining in their natal villages as adults and though many women will have relocated at the time of their marriage, this is a one-time event, and otherwise residential mobility (for example, for work or education) is relatively minimal. Similar assumptions of network stability have been previously made, as supported with partial longitudinal data, when conducting ALAAMs with other small-scale society data[59]. Future studies employing a longitudinal design may help avoid such ambiguity; however, such studies are likely to incur substantial time and labour costs unless able to harness previously collected data.

Another limitation is our reliance of direct questioning to assess FGMC preference and empirical and normative expectations regarding FGMC. This undoubtedly led to under-reporting of pro-FGMC preference[17] and possibly also led to both under-reporting of support and over-reporting of levels of social disapproval for the practice at the zone level. Though previous work has successfully used concealed questioning to elicit more honest FGMC preference reporting, such techniques produce population-level estimates of preference prevalence, while social network approaches necessitate individual-level data. Nevertheless, assuming under-reporting of personal preference suggests our models probably underestimate the strength of social influence and social selection. Such under- or over-reporting suggests the prevalence of pro-FGMC preference remains higher among this Arsi Oromo community than our figures indicate, leaving more girls at risk of being cut before marriage.

Finally, the laborious nature of social network data collection resulted in two constraints. First, we only collected networks from approximately half of the sample; while this allowed us to collect data from a greater number of kebele-zones and gain a better insight into pro-FGMC prevalence variation, it means not all known people named within the networks had the opportunity to report ties themselves. Despite our use of Bayesian techniques to estimate latent networks while accounting for common reporting errors and running a number of models with only the subsample of the network who could report, some bias inevitably remains in our estimates. Second, we were constrained to collecting only four network types (chatting, respect, money borrowing and marriage advice). The network types were chosen due to their mapping relationships either known to be important for cultural transmission in general or suggested to be important for FGMC, combined with our existing understanding of FGMC in this Arsi Oromo context. While this has allowed us to show the multiplicity of independent transmission paths, these relationships reflect only a small section of the myriad of relationships individuals hold. It is possible that social influence maintains FGMC across other types of network. Additionally, social selection on the basis of other, non-FGMC-related characteristics undoubtedly causes certain people to be more likely to socially interact, thus influencing the patterns of FGMC preference contagion. Further, we have only collected data from one of the many communities within which Arsi Oromo reside, thus it remains an open empirical question as to whether these findings generalize to the Arsi Oromo as a whole. Future work with the Arsi Oromo might productively both explore preference clustering among other network types and repeat our analyses in other communities, while work in other cultural contexts should be guided by existing contextual knowledge of FGMC.

To conclude, we find evidence within an Arsi Oromo community in south-central Ethiopia that pro-FGMC preference appears to persist at low levels through transmission via social influence or 'contagion' among people who chat together socially, within respect relationships and with whom the potential for money borrowing exists. We find no evidence that relatives are particularly socially influential regarding FGMC preference or that people would preferentially seek advice regarding the marriage of daughters from those with whom they share the same FGMC preference, suggesting the practice is not reinforced during marriage planning. Further, we find no support for the influential ideas that FGMC is a social coordination norm or persists due to the expectation that most other people want FGMC and/or approve of cutting. While we cannot be sure how generalizable these findings are, they nevertheless highlight that popular 'tipping point' intervention strategies would be ineffectual in this context and caution against such a one-size-fits-all approach. Diverse and locally informed strategies, limiting contagion on exposure to pro-FGMC attitudes, are needed needed when pursuing eradication efforts to end FGMC.

## Methods

### Data collection

Data collection took place between 2021 and 2022. Data were collected from nine neighbouring administrative kebele-zones, distributed across three kebeles (wards), all within the same woreda (district),

Hitosa, in the Arsi zone of Oromia. Kebele-zones were selected due to their high ethnic homogeneity, being predominantly ethnically Arsi Oromo, known to practice FGMC and geographical contiguity (Fig. 1a). Interviews were conducted in Afaan Oromoo by research assistants trained in demographic field survey methods, recruited and trained by E.G. at Addis Ababa University. Ethical and research permissions for the study were obtained by local and national authorities in Ethiopia, Addis Ababa University's Institute of Development and Policy Research (ref. IDPR/LT-001/2020) and from the University of Bristol (ref. 97942) in the UK. Informed consent was obtained from all participants.

A Household Census was initially conducted in 2021, with the aim of visiting each household within the nine kebele-zones, identifying all occupants aged 15 years or over (referred to as adults from now on) and assigning them individual ID codes (for further information, see the Supplementary Information). The final data reflect information regarding 5,578 individuals, across 1,949 households.

The Norms and Networks Survey was subsequently conducted in 2021–2022, with the aim of collecting preferences and expectations regarding FGMC from all individuals identified in the Household Census. The Norms and Networks Survey had four variants (1A, 1B, 2A and 2B); in all versions, respondents were asked to report their age, educational attainment, gender, the name of the head of their household and their relationship to them and questions about FGMC, and in versions 1A and 2A respondents were asked to report their social network connections (up to a maximum of ten) in four different domains (chatting, respect, money borrowing and marriage advice). For each alter (that is, social tie) named, their age, gender, relationship to the interviewee and whether they resided in the same kebele-zone was also recorded. Sixteen interviewers worked across the zones, with each interviewer responsible for specific households. Each interviewer was given a set of surveys ordered in a repeating pattern of 1A, 1B, 2A and 2B and instructed to administer the surveys in this order to participants as they were available; as such, social network data was collected semi-randomly, within and between households, from approximately half of participants in each kebele-zone who were broadly representative of the full sample (Supplementary Table 32). Where possible, the survey respondents and any alters named in respondents' social networks were matched against the Household Census to assign individual IDs (for further information on this process, see the Supplementary Information). In total, 5,181 interviews were conducted and the final sample encompasses 5,163 participants (including 50 who were not captured by the Household Census), 2,545 (49.3%) of whom reported social network data; two cases were excluded due to irreconcilable recording errors and 16 exclusions were made to remove one set of responses where individuals were interviewed twice. Of 54,632 alters named by respondents in response to six name generator questions, 41,235 (75.5%) were assigned IDs from the Household Census and 39,572 (72.4%) were found to have participated in the Norms and Networks Survey and are included in the following analyses.

Data and code for replicating the analyses are available on the Open Science Framework[60].

## Measures

**Social networks.** Respondents were asked to list up to ten names in response to the following name generators (additional wording delivered with the prompt can be seen in the Supplementary Information):

- Chatting network: 'Who do you spend time chatting with'
- Respect network: 'Who do you respect and admire?'
- Money network: 'From whom would you feel comfortable asking to borrow 100 birr if you needed it?' (referred to as 'borrow out' when inspected in isolation) and 'Who do you think would be comfortable asking to borrow 100 birr from you if they needed it?' (referred to as 'borrow in' when inspected in isolation). At the time of survey, 35 birr approximated US$1 and 100

birr would be enough to buy a week's worth of basic food items, that is, a substantial but not impossible amount of money to ask of someone in this economic context
- Advice network: 'Who would you go to for advice on preparing your daughter for marriage? Or if you do not have a daughter, who would you go to if you did have a daughter' (referred to as 'advice out' when inspected in isolation) and 'Who would come to you for advice on preparing their daughter for marriage?' (referred to as 'advice in' when inspected in isolation)

The double-sampled nature of the advice and money networks necessitates estimation of the latent networks to unite the elicited responses. To estimate the full across-zone networks we used the R package VIMuRe[19]. VIMuRe uses Bayesian inference to fit a latent network model from multiply reported network data, returning a posterior distribution of the latent variable, $\rho$, from which a point estimate for network is extracted. VIMuRe weights estimates for a potential tie between any two members of a whole network depending on whether both individuals in a given potential tie reported it. However, VIMuRe assumes both members of the tie were able to report, which is not the case in our sampling procedure, so we run estimation based on the full sample of Norms and Networks Survey respondents and the subsample of just those who reported their networks. In both sets of models, the direction of the effects are the same though the effect size is smaller in the former, thus we report the full sample in the main text and the subsample in the Supplementary Information. By estimating a latent network, we explicitly assume that our measurements of the network may be somewhat biased (for example, individuals may forget to report a tie) and are able to reconstruct our network of interest in a more principled and accurate way than, for instance, using deterministic rules (for example, taking the union or the intersection of nominations (reviewed in ref. 20)). We specified the default, weakly informative priors and, following De Bacco et al.[19], use the network mutuality ($\eta_{est}$) estimated by VIMuRe to set the threshold ($t\rho$) for a tie between alters at $t_\rho = 0.33 \times \eta_{est} + 0.10$.

**FGMC.** FGMC preferences were assessed by showing participants a set of five cards each depicting either FGMC, early marriage, work in the city, going to college or living close to home and asking whether they would want each option for a hypothetical daughter and then a hypothetical son's wife[17]. A respondent is considered to hold pro-FGMC preference if they responded 'yes' to either wanting FGMC for their daughter or daughter-in-law and anti-FGMC if they responded 'no' to both (pro- and anti-FGMC is used as shorthand throughout the methods and results section). While combining responses increases the number of pro-FGMC individuals than would be in either category if treated separately (see above), previous research using indirect questioning in a neighbouring community found no difference dependent on whether the daughter or daughter-in-law was the focus[17]. We frame wanting FGMC as a hypothetical parenting decision as it is the views of the parental generation that determine whether cutting occurs, with parents arranging and paying for cutting and the bride-groom's parents supplying food and drink for the ceremony[16]. As younger adults will not have been required to make the decision to cut a daughter yet and parents who cut their elder married daughters may or may not decide to cut younger yet to be married ones, this allows us to capture likely future behaviour.

To assess empirical expectations regarding pro-FGMC sentiment at the zone-level, we asked respondents to report what they perceived the prevalence of pro-FGMC sentiment to be by indicating on a visual scale depicting 0–100% in 10% increments how many men they thought would want FGMC for their daughter in their zone and then how many women.

To assess normative expectations regarding FGMC, respondents were also asked how they thought other people in their zone would respond 'if a family in this zone arranged female genital mutilation

for their daughter' with the response options 'approve of their action', 'disapprove of their action' and 'think it was none of their business'.

**Demographics.** Respondents were asked their age in years; responses to this question are approximate, as years of age is not something that is well tracked in this context, particularly among older community members, and responses between the two surveys for the same individual often varied.

Respondents were asked how many years of education they had completed; however, where individuals had gone beyond school, their qualifications were recorded instead; as a result, responses were recategorized into 'none', 'some primary', 'completed primary' or 'some secondary or beyond', with years mapped to the Ethiopian system of two 4-year cycles of primary schooling, followed by two 2-year cycles of secondary (as was the case at the time of survey).

Interviewers recorded the gender identification they inferred the respondent to hold (man or woman).

Respondents were asked to report whether they held a community role, and presented with the options 'traditional birth attendant', 'kebele leader', 'militia', 'teacher', 'religious leader' and 'other', with those reporting 'other' asked to specify what. Given the low absolute number of respondents reporting any role, the most common being teacher ($n = 99$) and birth attendant ($n = 91$), the responses are collapsed into 'role' or 'no role' (with 'no role' including four respondents recorded as 'other' but not having specified what).

Respondents were asked to report their religion, with the response options 'Muslim', 'Orthodox', 'Protestant', 'Waqeffatta' and 'Other'.

The kebele-zone in which individuals were censused are relabelled 1–9 for the purposes of anonymization, in order of their pro-FGMC preference prevalence (1 being the highest). A kebele-zone is a subdivision within a kebele (that is, the smallest administrative unit in the Ethiopian governance system), distinct from a zone (that is, the third administrative structure next to the regional national state within the federal system). As part of the Household Census, the GPS location was recorded, this information is used for visualization purposes only here (Fig. 1a) to give a sense of the geographic distribution of households across which our network measures spread, with longitude and latitude values redacted in the figure and dataset.

### Analytical strategy

First, we calculated Jaccard similarity coefficients for all network pair combinations to explore the possibility that our six name generators are eliciting the same latent network, potentially a kin network that could not be directly mapped due to socioecological logistical constraints (that is, patrilocality means women disperse on marriage, often substantial distances).

For each of our networks, we report the following measures of average centrality, estimated using the igraph package[61], dependent of whether individuals were pro- or anti-FGMC: in degree, that is, the number of nominations received; out degree, that is, the number of nominations made; vertex betweenness, which measures the extent to which an individual lies on the path between other individuals in the network, proxying their control over the flow of information between other; and harmonic centrality, which measures how close an individual is to all other individuals in the network and proxies the speed at which information could spread from a given individual to all other individuals. Higher scores in each of these measures indicates greater centrality. As measures of centrality are known to be biased by partial network sampling[62], we do not interpret the resulting values directly; however, to the extent that the networks of pro- and anti-individuals can be assumed to be similarly impacted, their comparison is of interest.

**Social influence.** To test for signals of simple or 'direct' social contagion of pro-FGMC preference, indicative of social influence, we ran a separate ALAAM for each network type using the BayesALAAM

function, available as part of MultivarALAAMalt.R[10]. The outcome variable, $Y$, is a binary measure of whether an individual holds anti- (0) or pro-FGMC (1) preference. A direct contagion model assumes partial conditional dependence, such that any two individuals (for example, individual $i$ and individual $j$) outcome variables (for example, $Y_i$ and $Y_j$) are conditionally dependent if and only if they are connected by a tie in a network, $X$, for example, when $X_{ij} = 1$ (ref. 10). We specified the default, minimally informative priors recommended by Koskinen and Daraganova[10], which ensure that our priors are informative enough to stabilize estimation without imposing overly strong assumptions on the parameter space[63]. Posteriors were estimated with Markov chain Monte Carlo as described in Koskinen and Daraganova[10]. The predictor variable of interest is then the direct contagion measure $\sum_{i<j} Y_i Y_j X_{ij}$, based on ties in the chatting, respect and latent money-borrowing networks, with a positively skewed posterior distribution indicating a respondent is more likely to be pro-FGMC if they are directly connected to another respondent who holds pro-FGMC preference. The statistical principle of hierarchy requires that out degree (that is, the number of nominations sent by a respondent) is also controlled for, as the lower-order effect of the contagion measure. In all models we then include age, gender, education, kebele-zone and in degree (that is, the number of nominations received); the conceptual directed acyclic graph informing this adjustment set is shown in Supplementary Fig. 1. To explore the possibility that contagion occurs differentially depending on whom ties are shared with, we also re-ran each model assessing kin and non-kin only ties. We also conducted post hoc explorations of models with the inclusion of additional dependencies, allowing for the estimation of other forms of contagion as facilitated by the BayesALAAM package (reciprocal, indirect, closed indirect and transitive contagion; see the Supplementary Information for definitions). Goodness-of-fit diagnostics for all models can be found in Supplementary Tables 11–19; as these models added little inferential insight beyond that of the simpler direct contagion models, we present the original models here and the posterior distributions for the 'best fitting' post hoc models can be found in Supplementary Fig. 8 (the output for the remaining models and model diagnostics can be found on the Open Science Framework[60]).

**Social selection.** To test for signals of social selection in the advice network based on shared FGMC preference, we ran combined stochastic block and social relations models using the R package STRAND[20]. The outcome variable is the existence of a tie (0/1) within the matrices of either out-going or in-coming ties reflecting whether a given individual would go to each of the other members of the whole network for advice. STRAND's Bayesian latent network model function estimates the latent network (avoiding the need for pre-estimation with VIMuRe), estimating the probability of a tie using both reporting patterns and the network members' characteristics entered as parameters within the model—and adjusting for typical biases associated with double-sampled, self-report network data. The parameters used to predict ties within the model are grouped into four different types of effect: block effects include FGMC preference and gender and assess the propensity of an individual to nominate those who share an attribute; focal effects include age, education and community role, and assess the propensity of a focal individual to nominate alters based on the focal individual's characteristics; target effects also include age, education and community role, and assess the propensity of a given individual to be nominated based on their characteristics; dyad effects include ties in the chatting, respect and latent money-borrowing (estimated with VIMuRe) networks and assess the propensity of a tie based on the relationship between any two individuals. We specified the default, weakly informative priors recommended in ref. 20. Posteriors were estimated using Markov chain Monte Carlo as implemented in CmdStanR v.2.34.1 (ref. 64). STRAND currently cannot adjust for biases introduced by our partial network sampling; thus, we confine

our analyses to potential ties between those who did report their networks. Furthermore, as STRAND models are estimating a host of individual-level and dyadic-level random effects, the computational power required for estimation grows exponentially as sample size increases. For this reason, we are constrained to running separate models for each kebele-zone and estimating the probability of ties between members of the same kebele-zone; for a comparison of the ties retained versus those excluded see Supplementary Table 4, and to implementing a limited number of iterations and chains. For further details see the Supplementary Information and for model diagnostics see ref. 60.

### Reporting summary

Further information on research design is available in the Nature Portfolio Reporting Summary linked to this article.

## Data availability

The data are available via the Open Science Framework at https://osf.io/765vg/ (ref. 60).

## Code availability

The code is available via the Open Science Framework at https://osf.io/765vg/ (ref. 60).

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

## Acknowledgements

This project was funded by the British Academy's Heritage, Dignity and Violence Programme (award reference HDV190133) to M.A.G., E.G. and A.A. The funders had no role in study design, data collection and analysis, decision to publish or preparation of the manuscript. We would like to thank the members of the Arsi Oromo communities who took part in study, the research assistants who conducted the surveys, and members of the Bristol Anthropology of Public Health Group and the LSHTM Evolutionary Demography Research Group who commented on an early draft of the paper.

## Author contributions

S.M. refined the research questions, curated the data, led the analytical strategy, performed the analyses and wrote the first manuscript draft. E.G. conceived the project, secured funding, consulted on data collection protocols and led fieldwork and data collection. A.A. conceived the project, secured funding, consulted on data collection protocols and consulted on the analytical strategy. D.R. consulted on the analytical strategy and assisted with analyses. J.A.H. led the design of data collection protocols. M.A.G. conceived and coordinated the project, secured funding, consulted on data collection protocols and consulted on the analytical strategy. All authors contributed to manuscript revisions and editing.

## Competing interests

The authors declare no competing interests.

## Additional information

**Correspondence and requests for materials** should be addressed to Sarah Myers or Mhairi A. Gibson.

# Reporting Summary

## Statistics

For all statistical analyses, confirm that the following items are present in the figure legend, table legend, main text, or Methods section.

| n/a | Confirmed | |
|---|---|---|
| ☐ | ☒ | The exact sample size (*n*) for each experimental group/condition, given as a discrete number and unit of measurement |
| ☐ | ☒ | A statement on whether measurements were taken from distinct samples or whether the same sample was measured repeatedly |
| ☐ | ☒ | The statistical test(s) used AND whether they are one- or two-sided *Only common tests should be described solely by name; describe more complex techniques in the Methods section.* |
| ☐ | ☒ | A description of all covariates tested |
| ☐ | ☒ | A description of any assumptions or corrections, such as tests of normality and adjustment for multiple comparisons |
| ☐ | ☒ | A full description of the statistical parameters including central tendency (e.g. means) or other basic estimates (e.g. regression coefficient) AND variation (e.g. standard deviation) or associated estimates of uncertainty (e.g. confidence intervals) |
| ☒ | ☐ | For null hypothesis testing, the test statistic (e.g. *F*, *t*, *r*) with confidence intervals, effect sizes, degrees of freedom and *P* value noted *Give P values as exact values whenever suitable.* |
| ☐ | ☒ | For Bayesian analysis, information on the choice of priors and Markov chain Monte Carlo settings |
| ☐ | ☒ | For hierarchical and complex designs, identification of the appropriate level for tests and full reporting of outcomes |
| ☒ | ☐ | Estimates of effect sizes (e.g. Cohen's *d*, Pearson's *r*), indicating how they were calculated |

*Our web collection on statistics for biologists contains articles on many of the points above.*

## Software and code

Policy information about availability of computer code

| Data collection | No software used, papser surveys only. |
|---|---|
| Data analysis | Data analysis was conducted using R Studio version 4.1.3; social selection modelling was conducted using the package STRAND version 0.0.0.9000; social influence modelling was conducted with the BayesALAAM function, part of MultivarALAAMalt.R available on GitHub and downloaded on 23/08/2023. All code used is available on the Open Science Framework at https://osf.io/765vg/. |

For manuscripts utilizing custom algorithms or software that are central to the research but not yet described in published literature, software must be made available to editors and reviewers. We strongly encourage code deposition in a community repository (e.g. GitHub). See the Nature Portfolio guidelines for submitting code & software for further information.

## Data

Policy information about availability of data

All manuscripts must include a data availability statement. This statement should provide the following information, where applicable:
- Accession codes, unique identifiers, or web links for publicly available datasets
- A description of any restrictions on data availability
- For clinical datasets or third party data, please ensure that the statement adheres to our policy

The data is available on the Open Science Framework at https://osf.io/765vg/.

# Research involving human participants, their data, or biological material

Policy information about studies with <u>human participants or human data</u>. See also policy information about <u>sex, gender (identity/presentation), and sexual orientation</u> and <u>race, ethnicity and racism</u>.

| | |
|---|---|
| Reporting on sex and gender | Throughout the manuscript we refer to the gender of survey respondents, as this is the most accurate terminology by academic standards. This information was inferred by field assistants conducting interviews, so as not to cause offense to respondents (a risk in this study context). However, the study community socially recognises only two genders, drawing no distinction between the concepts of sex and gender, and local field assistants originally coding the data recorded a variable 'sex' with options 'male' or 'female'. For the sake of transparency, we have retained this wording, altering the terminology during the model coding and signposting where this occurs in the code notation.<br>The sample is composed of 49.8% men and 50.2% women.<br>As gender likely plays a role in the development of thoughts and feelings regarding female genital mutilation/cutting, various outcomes of interest are reported disaggregated by gender and it is included as a parameter in our models. |
| Reporting on race, ethnicity, or other socially relevant groupings | The study communities were selected for the ethnic homogeneity (Arsi Oromo), which is of relevance when a cultural behaviour such as female genital mutilation/cutting. During the review process, information regarding self-reported religious affiliation was added, serving only to reinforce the relative homogeneity of the sample: Muslim 94.1%, Orthodox Christian 5.8%, other 0.1%. Neither ethnicity or religion are included within our analyses.<br>Respondents are grouped by self-reported educational attainment and community role.<br>We use directed acyclic graphs to determine the variables to include in our models to minimise confounding. |
| Population characteristics | N = 5163<br>Gender: men 49.8%, women 50.2%.<br>Age: median 30 years, IQR 25 years, range 15 - 99 years (note age is a self-reported approximation in this context).<br>Education: none 24.2%, some primary 46.8%, completed primary 11.4%, some secondary or beyond 17.7%.<br>Community role: none 92.5%, role 7.6%.<br>FGMC preference: pro-FGMC 6.3%, anti-FGMC 93.7%. |
| Recruitment | Every household in the study area, identified using government lists, was initially censused; household's were then visited again and invited to take part in the main study |
| Ethics oversight | University of Bristol (UK) and Addis Ababa University (Ethiopia). |

Note that full information on the approval of the study protocol must also be provided in the manuscript.

# Field-specific reporting

Please select the one below that is the best fit for your research. If you are not sure, read the appropriate sections before making your selection.

☐ Life sciences ☒ Behavioural & social sciences ☐ Ecological, evolutionary & environmental sciences

For a reference copy of the document with all sections, see <u>nature.com/documents/nr-reporting-summary-flat.pdf</u>

# Behavioural & social sciences study design

All studies must disclose on these points even when the disclosure is negative.

| | |
|---|---|
| Study description | The study is a quantitative, cross-sectional, observational study. |
| Research sample | The research sample approximates the entire adult population (individuals aged 15 years or over) of 9 neighbouring kebele-zones (N = 5163), thus is representative of these communities. Kebele-zones were selected due to their high ethnic homogeneity, being predominantly ethnically Arsi Oromo, known to practice female genital mutilation/cutting (FGMC), and geographical contiguity. |
| Sampling strategy | The entire population was targeted for inclusion |
| Data collection | Data collection took place between 2021 and 2022. Data was collected from nine neighbouring administrative kebele-zones (rural villages), distributed across three neighbouring Kebeles (sub districts) in South Central Ethiopia. Kebele-zones were selected due to their being predominantly ethnically Arsi Oromo, known to practice female genital mutilation (FGMC), and geographical contiguity. Data was recorded using pen and paper. Surveys were conducted in Afan Oromo by research assistants trained in demographic field survey methods, recruited and trained by EG at Addis Ababa University. |
| Timing | Surveys were undertaken 2021-2022. |
| Data exclusions | Data from 18 interviews were excluded from analyses: 2 cases were excluded due to irreconcilable recording errors and 16 exclusions were made to remove one set of responses where individuals were interviewed twice. |

| | |
|---|---|
| Non-participation | Our household census recorded 5578 adults residing within our study population, of whom 5165 (92.6%) took part in the study. Reasons for non-participation were not systematically documented but include uncontactability. |
| Randomization | Participants were not allocated into experimental groups. |

# Reporting for specific materials, systems and methods

We require information from authors about some types of materials, experimental systems and methods used in many studies. Here, indicate whether each material, system or method listed is relevant to your study. If you are not sure if a list item applies to your research, read the appropriate section before selecting a response.

## Materials & experimental systems

| n/a | Involved in the study |
|---|---|
| ☒ | ☐ Antibodies |
| ☒ | ☐ Eukaryotic cell lines |
| ☒ | ☐ Palaeontology and archaeology |
| ☒ | ☐ Animals and other organisms |
| ☒ | ☐ Clinical data |
| ☒ | ☐ Dual use research of concern |
| ☒ | ☐ Plants |

## Methods

| n/a | Involved in the study |
|---|---|
| ☒ | ☐ ChIP-seq |
| ☒ | ☐ Flow cytometry |
| ☒ | ☐ MRI-based neuroimaging |

## Plants

| | |
|---|---|
| Seed stocks | *Report on the source of all seed stocks or other plant material used. If applicable, state the seed stock centre and catalogue number. If plant specimens were collected from the field, describe the collection location, date and sampling procedures.* |
| Novel plant genotypes | *Describe the methods by which all novel plant genotypes were produced. This includes those generated by transgenic approaches, gene editing, chemical/radiation-based mutagenesis and hybridization. For transgenic lines, describe the transformation method, the number of independent lines analyzed and the generation upon which experiments were performed. For gene-edited lines, describe the editor used, the endogenous sequence targeted for editing, the targeting guide RNA sequence (if applicable) and how the editor was applied.* |
| Authentication | *Describe any authentication procedures for each seed stock used or novel genotype generated. Describe any experiments used to assess the effect of a mutation and, where applicable, how potential secondary effects (e.g. second site T-DNA insertions, mosiacism, off-target gene editing) were examined.* |

