## [Peer Review File · Nature Human Behaviour]

Social clustering of preference for female genital mutilation/cutting (FGMC) in South-Central Ethiopia

Corresponding Author: Professor Mhairi Gibson

Version 0:

Decision Letter:

21st October 2024

Dear Professor Gibson,

Thank you once again for your manuscript, entitled "Social clustering of preference for female genital mutilation/cutting (FGMC) in South-Central Ethiopia," and for your patience during the peer review process.

Your manuscript has now been evaluated by 2 reviewers, whose comments are included at the end of this letter. Please note that we did not succeed in securing a technical reviewer in this round, but will invite one in the next round. Although the reviewers find your work to be of interest, they also raise some important concerns. We are interested in the possibility of publishing your study in Nature Human Behaviour, but would like to consider your response to these concerns in the form of a revised manuscript before we make a decision on publication.

To guide the scope of the revisions, the editors discuss the referee reports in detail within the team, including with the chief editor, with a view to (1) identifying key priorities that should be addressed in revision and (2) overruling referee requests that are deemed beyond the scope of the current study. We hope that you will find the prioritised set of referee points to be useful when revising your study. Please do not hesitate to get in touch if you would like to discuss these issues further.

1) Our reviewers ask that you provide more details and information about the study. Please provide information on context, demographics and prevalence of FGMC, as well as ethnic diversity and how you dealt with this in the current context.

2) Related to the previous point, our reviewers raise questions about the generalisability of the findings. Please carefully address this point and discuss potential limitations to the generalisability of your findings in your Discussion section.

In sum, we invite you to revise your manuscript taking into account all reviewer and editor comments. We are committed to providing a fair and constructive peer-review process. Do not hesitate to contact us if there are specific requests from the reviewers that you believe are technically impossible or unlikely to yield a meaningful outcome.

We hope to receive your revised manuscript within two months. I would be grateful if you could contact us as soon as possible if you foresee difficulties with meeting this target resubmission date.

- Include a "Response to the editors and reviewers" document detailing, point-by-point, how you addressed each editor and referee comment. If no action was taken to address a point, you must provide a compelling argument. When formatting this document, please respond to each reviewer comment individually, including the full text of the reviewer comment verbatim followed by your response to the individual point. This response will be used by the editors to evaluate your revision and sent back to the reviewers along with the revised manuscript.
- Highlight all changes made to your manuscript or provide us with a version that tracks changes.

Link Redacted

We look forward to seeing the revised manuscript and thank you for the opportunity to review your work. Please do not hesitate to contact me if you have any questions or would like to discuss these revisions further.

Sincerely,

[Redacted]

[Redacted]

[Redacted]

Nature Human Behaviour

Reviewer expertise:

Reviewer #1: FGMC in the Ethiopian context and the role of social networks

Reviewer #2: FGMC in general

REVIEWER COMMENTS:

Reviewer #1 (Remarks to the Author):

Manuscript: Social clustering of preference for female genital mutilation/cutting (FGMC) in South-Central Ethiopia

Thank you for giving me the opportunity to review this work. The manuscript is about estimating social influence and social selection on preference and expectations surrounding FGMC among Arsi Oromo adults in Ethiopia. The topic is an important public health issue in Ethiopia, particularly in the study area and the region (Oromia), backed by a range of reasons for its continuation. The study is well designed and explored many aspects perceived vital for the practice in its continuation. Authors found that FGMC preference is transmitting via social influence within chatting, respect, and money-borrowing networks, and most importantly, like relatives, non-relatives are also important for the preference of FGM/C, which shows that there could be another factor that gear the practice to perpetuate. The results are relevant in many fields including, but not limited to, public health, behavioural and social sciences. The findings have also potential to influence policy.

I only have some issues with the draft, which I think are important to be addressed before publication. I will present my comments/suggestions and concerns as follow:

Minor comments

Line 75: the Afaan Oromoo words/statements require editions, as in its current state do not stand for the meaning the authors intended. Change 'in Oromiffa' to 'in Oromiffaa' and change 'huuba irra fuudhuu' to huuba irraa fuudhuu'.

It would be important to give a brief description of the study area because administratively there are two Arsi zones with may be culturally and socially different Arsi Oromos, which may have implications for the problem under investigation (West and East Arsi). Which zone? Woredas? Kebeles? I can guess where this study area is in Arsi, but this is hard for readers and may be for implements that have no clue where this study is conducted.

It would be good to give a brief overview of why this study location is selected, while the problem is across the region (Oromia) and the country.

Additionally, it would be good to briefly mention the variation in age by which a female/daughter undergo the practice in different parts of the country (good to look at the recent UN report on this). This is performed at early age in some places and at the time of marriage in some places.

Major concerns

- It is not clear that where religion is falling among these networks (chatting, respect, and money-borrowing), people take religion as a core driver of the practice and intention for the continuation of FGM/C.

- Again, this work missed to discuss the role of religion in perpetuating the practice in Ethiopia and specifically in the study area. The practice has different justifications at different parts of the country, despite no clear demarcation between reasons, and religion is among the major reasons mentioned widely in Arsi and Bale. This should be discussed both in the

background and in discussion (limitation?) section of this work.

- Arsi, both the West and East, is known by multiple ethnic group occupants, despite Oromo is the pre-dominant one (as mentioned by the authors). So, the question here is, how researchers managed this variation in ethnic group in their sampling, given not everyone in the study area endorses the practice (i.e., there are ethnic groups in the study area who do not belong to Oromo ethnic group who may or may not accept the practice)? Addressing this is important as it has implications on the whole research (e.g., conclusion and recommendation).

- One could imagine how hard to implement this method in a wider area than the authors did now, but my concern is it may be questionable to conclude this for the whole Arsi Oromo with one out of may be more than 30 Woredas where Arsi Oromo resides. Things including age at FGM, reasons for FGM (e.g., social, religion) could be vary within the Arsi Oromo, so that should be in consideration for generalisation.

Reviewer #2 (Remarks to the Author):

This article uses rich network data collected among Arsi Oromo agropastoralists in Ethiopia to analyze how attitudes toward FGMC are correlated within different kinds of social networks. The authors leverage the different types of networks to distinguish between the effects of social selection and social selection on the clustering of attitudes. The authors find limited impacts of social selection within marriage advice networks and limited evidence for social coordination around FGMC, challenging theoretical frameworks commonly used to explain the persistence of FGMC.

I am not an expert in network methods. The methods are reasonably clearly described and seem appropriate to me. I focus my comments on framing and interpretation of results.

1. My biggest questions are about the interpretation of results within this specific context and, by extension, the degree to which these findings might be expected to be relevant to other settings and contexts.

(a) There is not much information provided about the Arsi Oromo agropastoralists among whom the study is carried out. It seems that these communities are ethnically homogenous, but this is not explicitly stated. There is no information provided about religion or about common forms of agricultural production (are people in these communities predominantly herders? Do they migrate with animals?). The article does note that FGMC generally takes place shortly before marriage. All of these factors are relevant for the way people think about FGMC and, thus, for understanding the contexts where these findings might be expected to generalize.

(b) My understanding from reading the article is that this is a setting where FGMC was previously very common but where prevalence and approval have been declining rapidly. (I'm basing this on number presented in different places reporting that 90% of married women interviewed in 2010 had been cut, 87% of their daughters age 35 and over, and 53% of daughters age 25 and under; in 2016 22% of the community favored the continuation of FGMC. I couldn't find any data on prevalence at the time of the survey analyzed in this paper.) To me, this seems like a success story in the effort to eliminate FGMC – prevalence has fallen a lot, and most people want it to continue to fall. Yet the article is framed as an effort to understand persistence. I take the authors' point that any level of approval can lead to resurgence, but I think focusing on persistence leads to some misleading conclusions. For example, the authors find that marriage advice networks are not highly selective on FGMC attitudes, and conclude that FGMC is not driven by marriage conventions. An alternative interpretation could be that the lack of selection in marriage advice networks is the result of the rapid decline in FGMC prevalence, or, perhaps, that the weakness of marriage conventions in this setting is one of the factors that facilitated rapid decline. I don't think it's appropriate to conclude, based on this setting, that marriage networks aren't important in general in maintaining FGMC prevalence. (I don't have strong opinions about the importance of marriage conventions in general; this is just one example of an alternative possible interpretation.)

2. Related to the previous, I think it's important to provide a clear statement about how high FGMC prevalence is among current cohorts of young women, the level of FGMC preferences, and how strongly correlated preferences are with behavior. Your data may not allow you to be fully specific on these issues – please be clear about what we know and what we don't know.

3. One of the limitations of social convention theory, I think, is that it's not very specific about predictions. I.e., convention theory predicts sudden, rapid decline – but how fast is “rapid”, and how low is “decline”? This lack of specificity poses a problem in testing the theory, and in particular in providing negative evidence. I think it is important to bring this up in discussing whether your results constitute evidence against convention theory.

4. The questions about whether FGMC should continue are asked both about (hypothetical) daughters and (hypothetical) daughters in law. I would think that preferences for daughters and preferences for daughters in law might be quite different, in ways that are relevant for understanding beliefs and practices. Can you provide some information on what responses look like for daughters vs daughters in law?

Minor point

5. p. 9 – you say that it seems unlikely that FGMC preference would be a strong selection factor shaping chatting, respect, or money-borrowing networks. I agree, but it seems possible that factors strongly correlated with FGMC preference could drive selection (eg, ethnicity, religion, other markers of social identity/social status). Maybe you control for these things, or maybe FGMC preferences aren't stratified along these lines. Can you provide a brief clarification?

Version 1:

Decision Letter:

4th February 2025

Dear Professor Gibson,

Thank you once again for your revised manuscript, entitled "Social clustering of preference for female genital mutilation/cutting (FGMC) in South-Central Ethiopia," and for your patience during the re-review process.

Your manuscript has now been evaluated by the same reviewers who evaluated your original manuscript, as well as a new reviewer with expertise in Bayesian auto logistic actor-attribute models (Reviewer 3). All reviewer feedback is included at the end of this letter. Although the reviewers found your manuscript to have improved during revision, they also raise some important outstanding concerns. We remain interested in the possibility of publishing your study in *Nature Human Behaviour*, but would like to consider your response to these outstanding concerns in the form of a revised manuscript before we make a decision on publication.

Reviewer 3 raises several important concerns regarding the Bayesian auto-logistic actor-attribute models. Additionally, they provide valuable guidance on how to address these issues. Please ensure that each of their concerns is thoroughly addressed with clear explanations and, where necessary, supporting evidence or methodological refinements.

Furthermore, as requested by Reviewer 2, please expand on the generalizability of your findings. This should include a discussion of the potential limitations, applicability to different contexts, and any assumptions that may impact the broader relevance of your results.

In sum, we invite you to revise your manuscript taking into account all reviewer and editor comments. We are committed to providing a fair and constructive peer-review process. Do not hesitate to contact us if there are specific requests from the reviewers that you believe are technically impossible or unlikely to yield a meaningful outcome.

We hope to receive your revised manuscript within 4-8 weeks. I would be grateful if you could contact us as soon as possible if you foresee difficulties with meeting this target resubmission date.

- Include a "Response to the editors and reviewers" document detailing, point-by-point, how you addressed each editor and referee comment. If no action was taken to address a point, you must provide a compelling argument. This response will be used by the editors and reviewers to evaluate your revision.
- Highlight all changes made to your manuscript or provide us with a version that tracks changes.

Link Redacted

We look forward to seeing the revised manuscript and thank you for the opportunity to review your work. Please do not hesitate to contact me if you have any questions or would like to discuss these revisions further.

Sincerely,

[Redacted Signature]

[Redacted Contact Information]

Reviewer expertise:

Reviewer #1: FGMC in the Ethiopian context and the role of social networks

Reviewer #2: FGMC in general

Reviewer #3: Bayesian auto logistic actor-attribute models (ALAAMs)

REVIEWER COMMENTS:

Reviewer #1 (Remarks to the Author):

I have no further comments.

Reviewer #1 (Remarks on code availability):

I have no expertise in R to review the provided code.

Reviewer #2 (Remarks to the Author):

You've addressed all the points from the previous set of reviews – I appreciate the additional information and clarification. In some places, the responses seem somewhat superficial. For example, you've provided information about how many people want FGMC for both their daughters and their daughters-in-law, how many for their daughters only, and how many for their daughters-in-law only. But in the main text, you don't provide the rationale for combining responses, which raises questions for the reader. (I see that this is provided in the methods section.)

The response regarding generalizability of the results is also unsatisfying. You've added in several places a phrase noting that this applies only to the specific community being studied. But surely you don't really think that there is no generalizable information to be drawn from studying this community – if so, what would be the point of publishing? In some places, you seem to suggest that this case study is useful as a negative example calling into question common theories. You could make that argument more explicit as a way of explaining to the reader what is useful about your study. Or, ideally, you would make a broader statement (speculative is fine) about what kinds of places results might be generalizable to and/or what kinds of places are different enough that these results might not apply.

(Related: in the discussion section, you underline the importance of deep cultural understanding in designing effective interventions. Of course it's true that all contexts are different and no interventions will be universally effective. But at the same time, it's not a feasible policy approach to conduct years-long research in every small sub-region before planning interventions. Policy makers might draw on local knowledge, but you also argue (reasonably, in my view) that the reasons people themselves give for a preference are not sufficient for a full understanding of the factors driving preferences (p. 11). So there needs to be some level of generalizable conclusion if you're going to argue that your results can inform policy.)

Minor comment: Where you're talking about differences across kebele zones, especially in FGMC attitudes which are a theoretically central outcome, it would be helpful to present confidence intervals or some kind of estimates of uncertainty in order to assess how meaningful these differences are. (Or maybe you don't think confidence intervals are appropriate since you have a full population?)

Reviewer #3 (Remarks to the Author):

The study is an attempt to understand the persistence of female genital mutilation/cutting (FGMC) in Ethiopia despite legal prohibitions and eradication efforts. The study tries to ascertain how social influence and social selection contribute to maintaining pro-FGMC norms.

The study is guided by three basic hypotheses:

1. Pro-FGMC preference spreads through social influence ("contagion") within social networks.
2. Social selection (homophily) plays a role in FGMC preference, with individuals seeking marriage advice from those who share their views.
3. FGMC is maintained as a social-coordination norm or due to empirical/normative expectations.

These hypotheses are tested on a sample of 5,163 adults from the Arsi Oromo community in Ethiopia. Network data are collected at a single point in time on four types of social relationships—chatting, respect, money-borrowing, and marriage-

advice networks.

- The study reports strong evidence of social influence in chatting, respect, and money-borrowing networks, suggesting pro-FGMC views spread through interpersonal ties. No evidence is reported of social selection in marriage-advice networks, implying individuals do not preferentially seek FGMC advice from like-minded people.
- The low prevalence (6.3%) of pro-FGMC preference contradicts the idea that FGMC is a social coordination norm. FGMC is not maintained by normative expectations—most respondents (89.7%) believe their community disapproves it.
- The study challenges dominant assumptions about FGMC persistence by showing that social influence, rather than coordination norms or social selection, plays a key role. It suggests that interventions should target social network dynamics rather than assuming community-wide coordination.

* * *

I understand that the manuscript has already been reviewed. What I received is a resubmission. The comments that the authors have received concentrate on issues that relate directly to the substantive problem that motivates the study, and on features of the empirical setting in which the study is situated. The authors seem to have taken the comments received on the earlier round to heart and have responded carefully to the concerns expressed by reviewers.

While I understand what is at stake in this research and I appreciate its importance, I am not a social anthropologist and I am unable to offer competent advice on the content of the study, or comment on the theoretical and policy implications of the results reported. Prior reviews focused on these aspects of the study.

My commentary will be brief and focused on the aspect of the study I can evaluate with sufficient competence. Specifically, my commentary will focus on the specification, estimation and evaluation of autologistic actor-attribute models (ALAAMs). I will not comment on the stochastic blockmodeling analysis used to rule out alternative explanations of the results based on social influence. This is because blockmodeling analysis appears only weakly related to the main objective of the paper - demonstrating that FGMC practices diffuse through social contagion. Moreover, using blockmodeling to rule out possibilities of social selection seems methodologically weak, as testing the significance of a blockmodeling simulation is statistically challenging.

* * *

Let me start by saying that I see the adoption on Bayesian variant of ALAAMs as unrelated to the purpose of the study and to the argument that supports it. More specifically, consider line 180:

“Therefore, we use Bayesian auto logistic actor-attribute models (ALAAMs) [10], which were developed to explore signals of social influence or ‘contagion’ within cross-sectional network data (. . .)”

Nothing of what is contained in the text prior to this statement (starting right below line 170) actually justifies the adverb “Therefore” that is used to suggest that the models is the logical consequence of the argument that is being developed.

The same argument holds for the text in line 150 which just highlights a generic feature of any statistical models for networks (social or otherwise), i.e., accounting for dependencies in the data. Nothing in the objective of the study (or in the data) justifies the adoption of a Bayesian framework. Furthermore, I should note that ALAAMs can now be estimated more efficiently, accurately, and reliably with the ALAMEE software (Stivala et al., 2024, PLOS Complex Systems). I note, additionally, that “priors” -- one of the distinctive features of Bayesian models -- are specified only for the purpose of stabilizing the estimates (line 1120) -- not as a strategy for producing more informative and reliable tests of the hypotheses orienting the study. A minor detail, to conclude: the “double-sampled name generators” strategy that is implemented is well-conceived and informative -- but perhaps less uncommon that the authors seem to believe -- at least in social network analysis (for an example, see Lusher et al., 2012 - Social Networks)

- Action point number 1: A more compelling and substantive argument is needed for justifying the adoption of Bayesian approach. At the moment, there is nothing inherently “Bayesian” in the way the authors think about their study, specify their models, and interpret their results.

* * *

Social contagion is the focus of the study. The contagion effect in ALAAMs has been derived to test the hypothesis of network autocorrelation -- a statistically significant tendency to observe connected nodes sharing an attribute.

But this form of social contagion is the simplest but not the only one possible. In fact, what distinguish ALAAMs from the network autocorrelation model is (see for example: Leenders, 2002 - Social Networks; Marsden & Friedkin, 1994- in Advances in social network analysis; Wang, Neuman, & Newman, 2014 - Social Networks; the earlier Doreian, Teuter, & Wang, 1984 - Sociological Methods & Research, and the more recent Butts, 2023 - arXiv preprint arXiv:2310.20163) is the possibility for discriminating among alternative theoretical mechanisms of social contagion. This is discussed in Parker et al., 2022 -- Organizational Research Methods. Prominent alternative mechanisms include, for example, structural equivalence (see, for example, Burt, 1987 -American Journal of Sociology, and Fujimoto & Valente, 2012- Social Science & Medicine), or other positions occupied in extra-dyadic local configurations like for example “Simmelian triads” (see, for example, Goh et al., 2014 - Small Group Research). Note that there are many forms of structural equivalence. Balance, for example, is one of them.

In brief: An ALAAM specified only with contagion and indegree and outdegree effects (Table S8-S10) are basically logit models for network autocorrelation. These models do not account for the complex network dependencies that are (correctly) presented as the main justification for adopting ALAAMs.

- Action point number 2: Use full analytical potential of the model. Contagion is a complex social phenomenon. Do not reduce it to one parameter. Exploit the full analytical potential of the ALAAM to reveal multiple, possibly competing, mechanisms of social contagion.

* * *

The dependent variable (the “attribute”) in ALAAMs is constrained to be binary. The dependent variable of the study (“whether an individual holds anti- (0) or pro-FGMC (1) preference”) seems to be appropriately defined. However, because ALAAMs are models for a single observation where the social network is fixed and used to predict the presence on attribute in connected nodes, it is particularly important that information on the node-specific attribute be collected *after* observing the network (this is explained in Parker et al. 2022. When networks and attributes are observed at the same time, simultaneity becomes an obstacle for an interpretation of the results in terms of social contagion – a process that requires time to operate.

- Action point number 3. Please clarify if network data and attribute data have been collected at the same exact time point. If they have, you will need to provide a stronger argument to support your decision to treat social networks as exogenous. This issue may be linked to the results of the blockmodeling analysis.

* * *

I was surprised not to find any attempt to establish how well the model fits the data (of course, I might have missed it. In this case, please ignore this comment and accept my apologies). ALAAMs share with ERGMs the difficulty of computing the likelihood – and hence to compute summary measures of goodness of fit. Like it is the case for ERGMs, the GOF of ALAAMs can be tested via simulation (see Parker et al., 2022 for a practical illustration. See Stivala et al., 2024 for a more technical discussion. The general simulation approach used to test the GOF of ALAAMs has been originally derived by Hunter, Goodreau, & Handcock, 2008 - Journal of the American Statistical Association in the context of ERGMs.

In the context of the current study, the goodness of fit should be assessed relative to that of a comparable logit model which represents the null model for ALAAMs. In particular, the ability of the full model (ALAAM) to predict the incidence of the attribute ($Y=1$ if pro-FGMC) in the sample should be compared with the prediction ability of the null model (logit). A didactic explanation may be found in Parker et al (2022).

- Action point number 4. Please examine how well the model fits the data by following best recommended practice. For ALAAMs and ERGMs well-understood simulation-based procedures have been developed and are available. The size of the networks may require special attention to computing time (and cost if CPU or GPU time is being metered and priced)

* * *

I enjoyed reading this work, and I am grateful for the opportunity to participate in the evaluation process. I hope my comments will contribute to refine the argument developed in the manuscript and help the authors to strengthen their work and increase its potential impact. I offered my comments only in this spirit.

References

- Burt, R. S. (1987). Social contagion and innovation: Cohesion versus structural equivalence. *American journal of Sociology*, 92(6), 1287-1335.
- Doreian, P., Teuter, K., & Wang, C. H. (1984). Network autocorrelation models: some Monte Carlo results. *Sociological Methods & Research*, 13(2), 155-200.
- Fujimoto, K., & Valente, T. W. (2012). Social network influences on adolescent substance use: disentangling structural equivalence from cohesion. *Social Science & Medicine*, 74(12), 1952-1960.
- Goh, K. T., Krackhardt, D., Weingart, L. R., & Koh, T. K. (2014). The role of simmelian friendship ties on retaliation within triads. *Small Group Research*, 45(5), 471-505.
- Hunter, D. R., Goodreau, S. M., & Handcock, M. S. (2008). Goodness of fit of social network models. *Journal of the American statistical association*, 103(481), 248-258.
- Leenders, R. T. A. (2002). Modeling social influence through network autocorrelation: constructing the weight matrix. *Social networks*, 24(1), 21-47.
- Lusher, D., Robins, G., Pattison, P. E., & Lomi, A. (2012). “Trust Me”: Differences in expressed and perceived trust relations in an organization. *Social Networks*, 34(4), 410-424.
- Marsden, P. V., & Friedkin, N. E. (1994). Network studies of social influence. In *Advances in social network analysis: Research in the social and behavioral sciences* (pp. 3-25). SAGE Publications, Inc..
- Parker, A., Pallotti, F., & Lomi, A. (2022). New network models for the analysis of social contagion in organizations: an introduction to autologistic actor attribute models. *Organizational Research Methods*, 25(3), 513-540.
- Stivala, A., Wang, P., & Lomi, A. (2024). ALAAMEE: Open-source software for fitting autologistic actor attribute models. *PLOS Complex Systems*, 1(4), e0000021.

Version 2:

Decision Letter:

Our ref: NATHUMBEHAV-24051775B

3rd April 2025

Dear Dr. Gibson,

Thank you for submitting your revised manuscript "Social clustering of preference for female genital mutilation/cutting (FGMC) in South-Central Ethiopia" (NATHUMBEHAV-24051775B). It has now been seen by the original referees and their comments are below. As you can see, the reviewers find that the paper has improved in revision. We will therefore be happy in principle to publish it in Nature Human Behaviour, pending minor revisions to satisfy the referees' final requests and to comply with our editorial and formatting guidelines.

We are now performing detailed checks on your paper and will send you a checklist detailing our editorial and formatting requirements within two weeks. Please do not upload the final materials and make any revisions until you receive this additional information from us.

Sincerely,

[REDACTED]

[REDACTED]

[REDACTED]

Nature Human Behaviour

Reviewer #4 (Remarks to the Author):

The reviewers have engaged with my comments and submitted an improved revision of their manuscript. I appreciate their efforts and I read their rejoinders with interest.

I hope to see their paper in print soon.

As the authors correctly note in their response, this paper is not an appropriate venue for opening statistical discussions.

However, I would invite the authors to read their own response carefully and focus on the potential inconsistency in the argument that their work is indifferent to the Bayesian framework they adopt, while at the same time mentioning the advantages of adopting it.

I am referring specifically to the formulation of priors which could play more a theoretical than a simple empirical role (as they do now).

But how theory is incorporated in informative priors is rarely made explicit in Bayesian models. Priors (typically in their diffuse form) are used as a computational or algorithmic stratagem to assist with convergence or reduce computational time (like what is now called ABC (approximate Bayesian Computation) originally based on the seminal work of Diggle and Gratton (JRSS(B), 1984) and, later, that of Tavaré, Balding and Griffith [Genetics, 1997]).

One merit of this paper is making the tension between theory and computation particularly clear (line 1190/95: "We specified the default, minimally informative priors . . . which ensure that our priors are informative enough to stabilize estimation without imposing overly strong assumptions on the parameter space).

To some this may sound like an opportunity lost to link assumptions about the structure of the parameter space to a well-crafted and informative theoretical argument.

Statisticians fear assumptions. Social scientists should not, because they can link them to theory. Statisticians fear restrictions to the parameter space. Social scientists look for restrictions that map onto interesting hypotheses. Like the authors try to do in this paper.

None of my comments should be taken as a criticism of this competent and interesting piece of work addressing an important problem in social and cultural anthropology. Perhaps just as an incentive for the authors to reflect on their own craft in future occasions that I hope will be many.

We would like to thank both reviewers for their insightful and constructive comments. We have made a number of alterations to our manuscript in response to each reviewer and consider it to be stronger as a result. The main changes consist of an increase in ethnographic detail and discussion of religion, as requested by both reviewers. We include a version of the manuscript with the changes highlighted and give a summary of our response to each specific reviewer comment below in blue.

Reviewer 1

Line 75: the Afaan Oromoo words/statements require editions, as in its current state do not stand for the meaning the authors intended. Change 'in Oromiffa' to 'in Oromiffaa' and change 'huuba irra fuudhuu' to huuba irraa fuudhuu'. Thank you for identifying this; we have made the latter two changes and replaced 'in Oromiffaa' to 'in Afaan Oromoo'.

It would be important to give a brief description of the study area because administratively there are two Arsi zones with may be culturally and socially different Arsi Oromos, which may have implications for the problem under investigation (West and East Arsi). Which zone? Woredas? Kebeles? I can guess where this study area is in Arsi, but this is hard for readers and may be for implements that have no clue where this study is conducted. We had purposefully kept our study site anonymous to protect our participants, but we appreciate the desire for more details from an interpretation and intervention perspective. We have now specified that the study site is in the Hitosa Woreda of the Arsi Zone (line 117). As we understand it, the former Arsi Zone is now split into West Arsi and Arsi, rather than East Arsi as suggested; therefore, we retain reference to Arsi as our study location.

It would be good to give a brief overview of why this study location is selected, while the problem is across the region (Oromia) and the country. This has now been added to the introduction (from line 110 and 847).

Additionally, it would be good to briefly mention the variation in age by which a female/daughter undergo the practice in different parts of the country (good to look at the recent UN report on this). This is performed at early age in some places and at the time of marriage in some places. This has now been added to the introduction (line 12).

It is not clear that where religion is falling among these networks (chatting, respect, and money-borrowing), people take religion as a core driver of the practice and intention for the continuation of FGM/C. We have now added details regarding the relative religious homogeneity of the sample to the manuscript (lines 284-289 and lines 299-309). With 94% of our sample identifying as Muslim, we are unable to draw conclusions with regards to whether practising Islam compared to another religion is a driver; however, pro-FGMC preference is not exclusive to Muslims in the sample and is expressed by Orthodox individuals too. In terms of the networks, while clustering by religion is apparent in the marriage advice network (see the figures below – for reference, the location of nodes are the same as those in Figure S5 (for marriage) and 1B (for chatting) where kebele-zone was indicated by colour instead), determining whether this clustering is driven by religion or kinship is challenging and somewhat beyond our scope. Clustering in the other networks by religion is less apparent (e.g. see the chatting network below). Adding religion to our social influence DAG does not alter the indicated control variables. While we do not use a DAG for our social selection model, as the appropriate way to do so is unclear, eight out of nine kebele-zones are highly religiously homogenous (>90% Muslim). Religious clustering appears to be confined to kebele-zone 7 (where 77% of the Orthodox individuals in the sample reside); since kebele-zone 7 is not an outlier in our

social selection models, we do not think the omission of religion is driving our null result in this location.

Marriage advice network – green = Muslim, orange = non-Muslim

Chatting network – green = Muslim, orange = non-Muslim

Again, this work missed to discuss the role of religion in perpetuating the practice in Ethiopia and specifically in the study area. The practice has different justifications at different parts of the country, despite no clear demarcation between reasons, and religion is among the major reasons mentioned widely in Arsi and Bale. This should be discussed both in the background and in discussion (limitation?) section of this work. *Though people may cite religion as a reason for continuing the practice, it is clear that *within* religious groups there is heterogeneity in FGMC preference. While we have not framed our paper explicitly as such, it is this heterogeneity which is of interest to us here and whether it can be predicted by social ties, rather than the specific justifications individuals who are pro-FGMC may give. We have now added this point to our discussion (lines 722-736).*

Arsi, both the West and East, is known by multiple ethnic group occupants, despite Oromo is the pre-dominant one (as mentioned by the authors). So, the question here is, how researchers managed this variation in ethnic group in their sampling, given not everyone in the study area endorses the practice (i.e., there are ethnic groups in the study area who do not belong to Oromo ethnic group who may or may not accept the practice)? Addressing this is important as it has implications on the whole research (e.g., conclusion and recommendation). *Our study site was selected for its ethnic homogeneity; though we did not ask people to state their ethnic group, the high proportion of Muslims (94%) in our sample is indicative of their being overwhelmingly Arsi Oromo. We have now clarified this in the text in both the introduction and methods sections (line 118 and lines 849-850).*

One could imagine how hard to implement this method in a wider area than the authors did now, but my concern is it may be questionable to conclude this for the whole Arsi Oromo with one out of may be more than 30 Woredas where Arsi Oromo resides. Things including age at FGM, reasons for FGM (e.g., social, religion) could be vary within the Arsi Oromo, so that should be in consideration for generalisation. *Indeed, it would be challenging to implement this kind of study to cover a*

representative sample of Arsi Oromo as a whole and agree this necessarily constrains how generalisable our findings are; we have altered our phrasing throughout the discussion to be more circumspect.

Reviewer 2

There is not much information provided about the Arsi Oromo agropastoralists among whom the study is carried out. It seems that these communities are ethnically homogenous, but this is not explicitly stated. There is no information provided about religion or about common forms of agricultural production (are people in these communities predominantly herders? Do they migrate with animals?). The article does note that FGMC generally takes place shortly before marriage. All of these factors are relevant for the way people think about FGMC and, thus, for understanding the contexts where these findings might be expected to generalize. We have now added more detail on these points to the introduction and results sections (lines 110-124 and lines 284-289).

My understanding from reading the article is that this is a setting where FGMC was previously very common but where prevalence and approval have been declining rapidly. (I'm basing this on numbers presented in different places reporting that 90% of married women interviewed in 2010 had been cut, 87% of their daughters age 35 and over, and 53% of daughters age 25 and under; in 2016 22% of the community favored the continuation of FGMC. I couldn't find any data on prevalence at the time of the survey analyzed in this paper.) The estimates noted here are the only ones we are aware of at this local level; the UNFPA report a figure of 46% of girls aged 15-19 cut in Oromia in 2016, but this encompasses a very large geographical area and multiple ethnic groups. We have added a statement to this effect in the discussion (lines 529-539).

To me, this seems like a success story in the effort to eliminate FGMC – prevalence has fallen a lot, and most people want it to continue to fall. Yet the article is framed as an effort to understand persistence. I take the authors' point that any level of approval can lead to resurgence, but I think focusing on persistence leads to some misleading conclusions. For example, the authors find that marriage advice networks are not highly selective on FGMC attitudes, and conclude that FGMC is not driven by marriage conventions. An alternative interpretation could be that the lack of selection in marriage advice networks is the result of the rapid decline in FGMC prevalence, or, perhaps, that the weakness of marriage conventions in this setting is one of the factors that facilitated rapid decline. I don't think it's appropriate to conclude, based on this setting, that marriage networks aren't important in general in maintaining FGMC prevalence. (I don't have strong opinions about the importance of marriage conventions in general; this is just one example of an alternative possible interpretation.) We would agree is a possibility and have now added a comment regarding shifts in marriage practices in the community which may be related (lines 558-563). Further, we did not mean to imply marriage networks in general aren't important and have altered our wording throughout the discussion to be clearer on the caution required when generalising beyond this community.

Related to the previous, I think it's important to provide a clear statement about how high FGMC prevalence is among current cohorts of young women, the level of FGMC preferences, and how strongly correlated preferences are with behavior. Your data may not allow you to be fully specific on these issues – please be clear about what we know and what we don't know. We have now added a statement regarding this in our discussion (lines 529-539); as we do not have a contemporary estimate for cutting it is not possible to know how strongly preferences and behaviour are correlated.

One of the limitations of social convention theory, I think, is that it's not very specific about predictions. I.e., convention theory predicts sudden, rapid decline – but how fast is “rapid”, and how low is “decline”? This lack of specificity poses a problem in testing the theory, and in particular in providing negative evidence. I think it is important to bring this up in discussing whether your results constitute evidence against convention theory. We would agree and have added a comment to this effect to our discussion (lines 605-609).

The questions about whether FGMC should continue are asked both about (hypothetical) daughters and (hypothetical) daughters in law. I would think that preferences for daughters and preferences for daughters in law might be quite different, in ways that are relevant for understanding beliefs and practices. Can you provide some information on what responses look like for daughters vs daughters in law? We have now added the descriptive statistics for these responses in our sample discussion and a brief comment on them in our methods section (lines 299-303 and lines 991-906).

You say that it seems unlikely that FGMC preference would be a strong selection factor shaping chatting, respect, or money-borrowing networks. I agree, but it seems possible that factors strongly correlated with FGMC preference could drive selection (eg, ethnicity, religion, other markers of social identity/social status). Maybe you control for these things, or maybe FGMC preferences aren't stratified along these lines. Can you provide a brief clarification? We have now more clearly highlighted this information in the main text (lines 304-309, lines 441-443, and lines 468-470). FGMC preference isn't strongly stratified by social or demographic marker. The clearest difference is by age, with older people being more likely to want FGMC. Age is also a marker of social status in this community and is controlled for in our models, as is community role – our other marker of social status. Ethnicity and religion are effectively synonymous within our sample and highly homogenous – we have added more information regarding this in the text (please also see our comments to reviewer 1 for more on our handling of religion).

Reviewer comments round 3

Dear Editor,

We would like to thank you for the opportunity to revise our manuscript. We also extend our thanks to the reviewers for their detailed comments, particularly reviewer 3 whose constructive advice has strengthened our analysis. To summarise, in response to reviewer 2 we have now rebalanced our discussion to highlight what generalisations we think can be made, without compromising on the caution previously requested by reviewer 1. In response to reviewer 3, we have now included both goodness-of-fit simulations and additional models exploring more complex forms of contagion in our supplementary information; as this amounts to 50 additional models, we include only the results from the best fitting models and provide the other models on our Open Science Framework page. We have also made changes to our phrasing in the introduction to place the emphasis on model-type, rather than the choice of Bayesian modelling. As it was not our intention to 'sell' Bayesian approaches, we do not think it necessary to justify our choice of Bayesian in the manuscript and to do so would distract from the purpose of our study; nevertheless, we have expanded on our rationale in our comments below.

We outline our responses to particular comments in blue and where relevant provide line references in the main manuscript.

REVIEWER COMMENTS:

Reviewer #1 (Remarks to the Author):

I have no further comments.

We thank reviewer 1 for taking the time to review our changes and happy they were found to be sufficient.

Reviewer #1 (Remarks on code availability):

I have no expertise in R to review the provided code.

Reviewer #2 (Remarks to the Author):

You've addressed all the points from the previous set of reviews – I appreciate the additional information and clarification. In some places, the responses seem somewhat superficial. For example, you've provided information about how many people want FGMC for both their daughters and their daughters-in-law, how many for their daughters only, and how many for their daughters-in-law only. But in the main text, you don't provide the rationale for combining responses, which raises questions for the reader. (I see that this is provided in the methods section.)

We have now added the rationale to the introduction as well, where we introduce the definition of our measure [line 103-7].

The response regarding generalizability of the results is also unsatisfying. You've added in several places a phrase noting that this applies only to the specific community being studied. But surely you

don't really think that there is no generalizable information to be drawn from studying this community – if so, what would be the point of publishing? In some places, you seem to suggest that this case study is useful as a negative example calling into question common theories. You could make that argument more explicit as a way of explaining to the reader what is useful about your study. Or, ideally, you would make a broader statement (speculative is fine) about what kinds of places results might be generalizable to and/or what kinds of places are different enough that these results might not apply.

(Related: in the discussion section, you underline the importance of deep cultural understanding in designing effective interventions. Of course it's true that all contexts are different and no interventions will be universally effective. But at the same time, it's not a feasible policy approach to conduct years-long research in every small sub-region before planning interventions. Policy makers might draw on local knowledge, but you also argue (reasonably, in my view) that the reasons people themselves give for a preference are not sufficient for a full understanding of the factors driving preferences (p. 11). So there needs to be some level of generalizable conclusion if you're going to argue that your results can inform policy.)

We appreciate this and the previous comment, we agree with hindsight we may have overshot when trying to encourage cautious application in response to earlier reviewer comments. We have now rebalanced parts of our discussion to emphasise our data act as a useful negative example, but also points to the potential for social influence to maintain pro-FGMC, even at low levels. Where costly social network data cannot be afforded, measures designed to stop pro-FGMC preferences being caught on exposure are likely to be a safe investment in the absence of strong evidence for mass coordination [line 701-746].

Minor comment: Where you're talking about differences across kebele zones, especially in FGMC attitudes which are a theoretically central outcome, it would be helpful to present confidence intervals or some kind of estimates of uncertainty in order to assess how meaningful these differences are. (Or maybe you don't think confidence intervals are appropriate since you have a full population?)

We are inclined to think confidence intervals are not appropriate given our sample approximates the full adult population; nevertheless, as under-reporting is likely and may be influenced by local dynamics, we are hesitant to draw strong inferences from these differences. We have now noted this [line 302].

We would like to thank reviewer 2 for taking the time to consider our manuscript once again, their insights have undoubtedly helped to improve it.

Reviewer #3 (Remarks to the Author):

The study is an attempt to understand the persistence of female genital mutilation/cutting (FGMC) in Ethiopia despite legal prohibitions and eradication efforts. The study tries to ascertain how social influence and social selection contribute to maintaining pro-FGMC norms.

The study is guided by three basic hypotheses:

1. Pro-FGMC preference spreads through social influence ("contagion") within social networks.
2. Social selection (homophily) plays a role in FGMC preference, with individuals seeking marriage advice from those who share their views.
3. FGMC is maintained as a social-coordination norm or due to empirical/normative expectations.

These hypotheses are tested on a sample of 5,163 adults from the Arsi Oromo community in Ethiopia. Network data are collected at a single point in time on four types of social relationships—chatting, respect, money-borrowing, and marriage-advice networks.

- The study reports strong evidence of social influence in chatting, respect, and money-borrowing networks, suggesting pro-FGMC views spread through interpersonal ties. No evidence is reported of social selection in marriage-advice networks, implying individuals do not preferentially seek FGMC advice from like-minded people.
- The low prevalence (6.3%) of pro-FGMC preference contradicts the idea that FGMC is a social coordination norm. FGMC is not maintained by normative expectations—most respondents (89.7%) believe their community disapproves it.
- The study challenges dominant assumptions about FGMC persistence by showing that social influence, rather than coordination norms or social selection, plays a key role. It suggests that interventions should target social network dynamics rather than assuming community-wide coordination.

* * *

I understand that the manuscript has already been reviewed. What I received is a resubmission. The comments that the authors have received concentrate on issues that relate directly to the substantive problem that motivates the study, and on features of the empirical setting in which the study is situated. The authors seem to have taken the comments received on the earlier round to heart and have responded carefully to the concerns expressed by reviewers.

While I understand what is at stake in this research and I appreciate its importance, I am not a social anthropologist and I am unable to offer competent advice on the content of the study, or comment on the theoretical and policy implications of the results reported. Prior reviews focused on these aspects of the study.

My commentary will be brief and focused on the aspect of the study I can evaluate with sufficient competence. Specifically, my commentary will focus on the specification, estimation and evaluation of autologistic actor-attribute models (ALAAMs). I will not comment on the stochastic blockmodeling analysis used to rule out alternative explanations of the results based on social influence. This is because blockmodeling analysis appears only weakly related to the main objective of the paper - demonstrating that FGMC practices diffuse through social contagion. Moreover, using blockmodeling to rule out possibilities of social selection seems methodologically weak, as testing the significance of a blockmodeling simulation is statistically challenging.

* * *

Let me start by saying that I see the adoption on Bayesian variant of ALAAMs as unrelated to the purpose of the study and to the argument that supports it. More specifically, consider line 180: “Therefore, we use Bayesian auto logistic actor-attribute models (ALAAMs) [10], which were developed to explore signals of social influence or ‘contagion’ within cross-sectional network data (. . .)”

Nothing of what is contained in the text prior to this statement (starting right below line 170) actually justifies the adverb “Therefore” that is used to suggest that the models is the logical consequence of the argument that is being developed.

We appreciate this points, our intent had not been to justify the choice of Bayesian here, but rather the choice of an ALAAM; we have now altered our wording.

The same argument holds for the text in line 150 which just highlights a generic feature of any statistical models for networks (social or otherwise), i.e., accounting for dependencies in the data. Nothing in the objective of the study (or in the data) justifies the adoption of a Bayesian framework.

Furthermore, I should note that ALAAMs can now be estimated more efficiently, accurately, and reliably with the ALAMEE software (Stivala et al., 2024, PLOS Complex Systems).

We thank the reviewer for this reference and would certainly be interested in exploring the potential of this software in future work; however, as Stivala et al. was published in December 2024 and we submitted our manuscript in May 2024, we hope we can be forgiven for not having used it. We have now included reference to this work for readers interested in ALAAMs [line 190].

I note, additionally, that “priors” -- one of the distinctive features of Bayesian models – are specified only for the purpose of stabilizing the estimates (line 1120) – not as a strategy for producing more informative and reliable tests of the hypotheses orienting the study.

While priors do help stabilise estimates, their role extends beyond this. In Bayesian modelling, priors influence inference by incorporating domain knowledge and preventing overfitting or underfitting. The use of flat (uninformative) priors—making inference rely solely on the likelihood function, which are similar to frequentist estimation procedures—can lead to overfitting by allowing extreme parameter values to dominate posterior estimates. Conversely, overly strong priors may cause underfitting, masking meaningful variation in the data. Weakly informative or regularising priors (which are used here) strike a balance by constraining implausible values while still allowing the data to meaningfully inform the model. This contributes to producing more reliable and robust estimates (see McElreath, *Statistical Rethinking*, 2nd ed., 2020, Ch. 7). Therefore, priors are not merely a tool for stabilising estimates but play a crucial role in improving the informativeness and reliability of statistical conclusions.

More generally, the actual estimation procedure of Bayesian models is a key distinction from frequentist methods (not just the incorporation of priors), which does provide several advantages. Unlike frequentist approaches (e.g., MLE, MoM, OLS), which treat parameters as fixed and rely on long-run frequency interpretations, Bayesian inference provides full probability distributions that allow for direct probability statements about parameters.

A minor detail, to conclude: the “double-sampled name generators” strategy that is implemented is well-conceived and informative – but perhaps less uncommon than the authors seem to believe – at least in social network analysis (for an example, see Lusher et al., 2012 - Social Networks)

We had not meant to imply double sampled name generators were novel, rather the novel element is the technique we use for estimating the latent network they tap; we have now altered our wording to clarify [line 152].

- Action point number 1: A more compelling and substantive argument is needed for justifying the adoption of Bayesian approach. At the moment, there is nothing inherently “Bayesian” in the way the authors think about their study, specify their models, and interpret their results.

We appreciate opinions differ regarding frequentist vs. Bayesian approaches. It is not our intention to ‘sell’ a Bayesian approach. Given this, we have largely removed reference to Bayesian in our introduction, instead introducing it in our results section (where is it necessary to guide interpretation). We hope these tweaks to the wording suffice to make it clear it is the choice of

ALAAM and block models that stem from our data and research questions, rather than the choice of a Bayesian approach. We believe that it is beyond the scope of this manuscript to relay (or attempt to solve) debates and divisions within the field.

We do, however, respectfully disagree with the claim that there is nothing inherently Bayesian in our study. Our approach is explicitly Bayesian, as we specify Bayesian models with weakly informative priors, we use Bayesian estimation, and we interpret full posterior distributions rather than relying on point estimates or frequentist confidence intervals. These are fundamental aspects of Bayesian inference and directly shape how we model uncertainty and draw conclusions from our data.

Furthermore, we question why a justification for adopting Bayesian methods is necessary when frequentist approaches are rarely, if ever, required to be justified in the same way. Bayesian methods offer well-documented advantages, including more coherent uncertainty quantification and the ability to incorporate prior information where appropriate. While we acknowledge differing opinions on statistical paradigms, our choice is guided by the needs of our study rather than an ideological preference. Given that Bayesian inference is central to how we estimate and interpret our results, we see no reason to obscure this fact or treat it as requiring special defence.

In sum, we appreciate the feedback and have adjusted the manuscript to ensure clarity that we are not trying to 'sell' Bayesian approaches, but this is simply the approach that we used—an approach that the GoF tests, as requested by the reviewer, indicate appropriately reflect the data (see below). Yet, we do now make reference to Stivala et al. 2024 to guide readers to other approaches to estimating ALAAMs. We maintain that justifying Bayesian estimation should not be held to a higher standard than justifying frequentist approaches, and we believe the current framing accurately reflects our methodological choices.

* * *

Social contagion is the focus of the study. The contagion effect in ALAAMs has been derived to test the hypothesis of network autocorrelation – a statistically significant tendency to observe connected nodes sharing an attribute.

But this form of social contagion is the simplest but not the only one possible. In fact, what distinguish ALAAMs from the network autocorrelation model is (see for example: Leenders, 2002 - Social Networks; Marsden & Friedkin, 1994- in Advances in social network analysis; Wang, Neuman, & Newman, 2014 - Social Networks; the earlier Doreian, Teuter, & Wang, 1984 - Sociological Methods & Research, and the more recent Butts, 2023 - arXiv preprint arXiv:2310.20163) is the possibility for discriminating among alternative theoretical mechanisms of social contagion. This is discussed in Parker et al., 2022 – Organizational Research Methods. Prominent alternative mechanisms include, for example, structural equivalence (see, for example, Burt, 1987 -American Journal of Sociology, and Fujimoto & Valente, 2012- Social Science & Medicine), or other positions occupied in extra-dyadic local configurations like for example “Simmelian triads” (see, for example, Goh et al., 2014 - Small Group Research). Note that there are many forms of structural equivalence. Balance, for example, is one of them.

In brief: An ALAAM specified only with contagion and indegree and outdegree effects (Table S8-S10) are basically logit models for network autocorrelation. These models do not account for the complex network dependencies that are (correctly) presented as the main justification for adopting ALAAMs.

- Action point number 2: Use full analytical potential of the model. Contagion is a complex social phenomenon. Do not reduce it to one parameter. Exploit the full analytical potential of the ALAAM to reveal multiple, possibly competing, mechanisms of social contagion.

We thank the reviewer for this useful suggestion and have now explored a number of additional contagion formulations (50 additional models), and compare the outcome of goodness-of-fit simulations to identify the model best able to return the network structure for each of our networks. We were constrained in this exploration to the additional contagion formulations available in the BayesALAAM package (reciprocal, indirect, closed indirect, and transitive); while we appreciate that there are numerous forms of contagion that could conceivably be explored, we do not have strong theoretical reasons to expect particular forms other than direct contagion are important and the interpretation of any such effects would be *post hoc*. We run GoF simulations for all 59 models and present the output in the supplementary information [SI tables S11-19]; given the volume of additional models (and the poor fit on a number of them) we present only the results of the best fitting models [SI tables S20-22 and Figure S8] and signpost interested readers to our Open Science Framework page to see the rest (<https://osf.io/765vg/files/osfstorage#>). Broadly, the effect of direct contagion is robust to the inclusion of additional contingencies; GoF simulations indicate the direct contagion-only model fits best for 3 of 9 networks, while model diagnostic tests indicate the direct contagion-only models have similar or better convergence and larger effective sample sizes, thus we retain our original models in the main text and present the additional results in the supplementary information/on OSF. Interestingly, we find two instances of a compelling signal of indirect contagion: i) when only non-kin alters are considered in the money network, indirect contagion has a similar signal to direct contagion and; ii) when only kin alters are considered in the respect network the posteriors suggest the direct contagion signal we see in the simpler model was potentially driven by indirect contagion. We highlight this for interested readers but are uncertain as to the explanation underlying this finding [line 471-93].

* * *

The dependent variable (the “attribute”) in ALAAMs is constrained to be binary. The dependent variable of the study (“whether an individual holds anti- (0) or pro-FGMC (1) preference”) seems to be appropriately defined. However, because ALAAMs are models for a single observation where the social network is fixed and used to predict the presence on attribute in connected nodes, it is particularly important that information on the node-specific attribute be collected *after* observing the network (this is explained in Parker et al. 2022. When networks and attributes are observed at the same time, simultaneity becomes an obstacle for an interpretation of the results in terms of social contagion – a process that requires time to operate.

- Action point number 3. Please clarify if network data and attribute data have been collected at the same exact time point. If they have, you will need to provide a stronger argument to support your decision to treat social networks as exogenous. This issue may be linked to the results of the blockmodeling analysis.

Our attribute and network data were collected at the same time, we had intended this to be clear from our discussion of its cross-sectional nature in the introduction; however, we have now inserted some additional wording in the introduction to leave no doubt [line 132]. That contagion takes time

to operate cannot be disputed, and we had sought to make clear in the introduction that any inferences we draw rely on researcher assumptions; nevertheless, we have now strengthened the reiteration of our reliance on assumptions in the discussion [line 794-801]. We share the concerns regarding simultaneity and the assumption of a fixed network, which are general limitations of using cross-sectional data and models to analyse social influence/transmission.

Given our ethnographic understanding of the study context, we have strong reasons to believe that the networks in our data are relatively stable and have now expanded on this in our discussion [line 801-17]. Social ties in this setting are not highly volatile but rather likely persist over time, which supports the assumption that the network can be treated as exogenous for the purpose of our analysis, and our expectation that it is highly unlikely that participants would simultaneously change both their social ties and FGMC preference at the given point of observation. Nonetheless, we have tried to make the limitations of our study clear in the discussion (with direct reference to the need for future work to collect longitudinal data), yet time-lagged data do not exist on this topic and it was not feasible for us to collect it. However, we think this limitation is counter balanced by the fact that waiting for others to collect prospective data means even less well evidenced assumptions will remain unchallenged in the literature and continue to inform potentially misguided health interventions.

We also recognise the challenges of using network data collected at a previous time point, including uncertainty about what the optimal time-lag could be before attribute data collection. Collecting network data too far in advance risks capturing outdated network structure, violating equilibrium assumptions by allowing for significant network changes. Conversely, waiting too long after network observation before measuring attributes does not fully resolve simultaneity concerns, as the exact time window for social influence effects remains unknown. Given these trade-offs, simultaneous data collection does not necessarily introduce greater bias than using past network data—rather, it ensures that the observed network structure is as relevant as possible to the attributes we model.

Additionally, while simultaneity can complicate causal claims about social contagion, ALAAMs do not inherently require a strict causal interpretation. They estimate the association between network structure and attributes, which is meaningful even if the precise temporal sequence of influence is uncertain. Our results have been interpreted in this light, with thorough treatment of the limitations.

To clarify, we have explicitly stated the timing of network and attribute data collection in the manuscript and reinforce our justification for treating the network as exogenous. However, given the stability of social ties in our setting and the limitations of alternative approaches, we maintain that our modelling choices are reasonable and appropriate.

* * *

I was surprised not to find any attempt to establish how well the model fits the data (of course, I might have missed it. In this case, please ignore this comment and accept my apologies). ALAAMs share with ERGMs the difficulty of computing the likelihood – and hence to compute summary measures of goodness of fit. Like it is the case for ERGMs, the GOF of ALAAMs can be tested via simulation (see Parker et al., 2022 for a practical illustration. See Stivala et al., 2024 for a more technical discussion. The general simulation approach used to test the GOF of ALAAMs has been originally derived by Hunter, Goodreau, & Handcock, 2008 - Journal of the American Statistical Association in the context of ERGMS.

In the context of the current study, the goodness of fit should be assessed relative to that of a comparable logit model which represents the null model for ALAAMs. In particular, the ability of the full model (ALAAM) to predict the incidence of the attribute ($Y=1$ if pro-FGMC) in the sample should be compared with the prediction ability of the null model (logit). A didactic explanation may be found in Parker et al (2022).

- Action point number 4. Please examine how well the model fits the data by following best recommended practice. For ALAAMs and ERGMs well-understood simulation-based procedures have been developed and are available. The size of the networks may require special attention to computing time (and cost if CPU or GPU time is being metered and priced).

We have now added information regarding goodness-of-fit to our supplementary information [see SI tables S11-19]. Koskinen provides a function for simulating goodness of fit within a Bayesian framework, based on the simulation procedures used for ERGMs. From these simulations, we present the mean and the 90% highest posterior density interval of the distribution of expected values from a given model for comparison against the observed network statistics. Inferences regarding goodness of fit are based on the distance between the mean and the observed value, and the width of the HPDI, with better fit indicated by a smaller distance between the mean and the observation, the observation falling within the HPDI and, where this is the case, a narrower interval (we have added this explanation to the methods section of our SI too). Across all networks, our original model containing direct contagion was a better fit than the comparable logit model.

* * *

I enjoyed reading this work, and I am grateful for the opportunity to participate in the evaluation process. I hope my comments will contribute to refine the argument developed in the manuscript and help the authors to strengthen their work and increase its potential impact. I offered my comments only in this spirit.

We would like to sincerely thank the reviewer for taking the time to make such detailed and constructive comments. We hope that we have sufficiently taken them on board and consider the manuscript to be improved as a result.

References

- Burt, R. S. (1987). Social contagion and innovation: Cohesion versus structural equivalence. *American journal of Sociology*, 92(6), 1287-1335.
- Doreian, P., Teuter, K., & Wang, C. H. (1984). Network autocorrelation models: some Monte Carlo results. *Sociological Methods & Research*, 13(2), 155-200.
- Fujimoto, K., & Valente, T. W. (2012). Social network influences on adolescent substance use: disentangling structural equivalence from cohesion. *Social Science & Medicine*, 74(12), 1952-1960.
- Goh, K. T., Krackhardt, D., Weingart, L. R., & Koh, T. K. (2014). The role of simmelian friendship ties on retaliation within triads. *Small Group Research*, 45(5), 471-505.
- Hunter, D. R., Goodreau, S. M., & Handcock, M. S. (2008). Goodness of fit of social network models. *Journal of the American Statistical Association*, 103(481), 248-258.
- Leenders, R. T. A. (2002). Modeling social influence through network autocorrelation: constructing the weight matrix. *Social Networks*, 24(1), 21-47.
- Lusher, D., Robins, G., Pattison, P. E., & Lomi, A. (2012). "Trust Me": Differences in expressed and

perceived trust relations in an organization. *Social Networks*, 34(4), 410-424.

Marsden, P. V., & Friedkin, N. E. (1994). Network studies of social influence. In *Advances in social network analysis: Research in the social and behavioral sciences* (pp. 3-25). SAGE Publications, Inc..

Parker, A., Pallotti, F., & Lomi, A. (2022). New network models for the analysis of social contagion in organizations: an introduction to autologistic actor attribute models. *Organizational Research Methods*, 25(3), 513-540.

Stivala, A., Wang, P., & Lomi, A. (2024). ALAAMEE: Open-source software for fitting autologistic actor attribute models. *PLOS Complex Systems*, 1(4), e0000021.